# The Impact of On-Policy Parallelized Data Collection on Deep Reinforcement Learning Networks

Walter Mayor [* 1]   Johan Obando-Ceron [* 2 3]   Aaron Courville [2 3]   Pablo Samuel Castro [2 3 4]

## Abstract

The use of parallel actors for data collection has been an effective technique used in reinforcement learning (RL) algorithms. The manner in which data is collected in these algorithms, controlled via the number of parallel environments and the rollout length, induces a form of bias-variance trade-off; the number of training passes over the collected data, on the other hand, must strike a balance between sample efficiency and overfitting. We conduct an empirical analysis of these trade-offs on PPO, one of the most popular RL algorithms that uses parallel actors, and establish connections to network plasticity and, more generally, optimization stability. We examine its impact on network architectures, as well as the hyperparameter sensitivity when scaling data. Our analyses indicate that larger dataset sizes can increase final performance across a variety of settings, and that scaling parallel environments is more effective than increasing rollout lengths. These findings highlight the critical role of data collection strategies in improving agent performance.

## 1. Introduction

Reinforcement Learning (RL) is a promising framework for addressing a wide range of decision-making tasks, from robotics and autonomous driving to game playing and resource optimization (Schwarzer et al., 2023; Vinyals et al., 2019; Bellemare et al., 2020; Fawzi et al., 2022). Central to the success of RL is the iterative process of data collection and policy learning, wherein agents interact with an environment to gather experience, which is subsequently used to improve decision-making policies (Sutton & Barto, 2018).

This interaction forms the foundation of RL's ability to discover and refine strategies that can handle high-dimensional and dynamic decision spaces. However, many RL tasks, particularly those involving complex environments, suffer from a critical bottleneck: the need for extensive and diverse training data to effectively generalize across the state space (Cobbe et al., 2020). This bottleneck is further compounded in online RL, where data arrives sequentially over time in a non-stationary fashion (Khetarpal et al., 2022).

Traditional RL methods often rely on single-environment simulations which limits the speed and diversity of data collection. This approach, while straightforward, is inherently slow and can restrict the scope of exploration, leading to suboptimal policy learning (Rudin et al., 2022; Singla et al., 2024). To address these limitations, recent advances in parallelized simulations, powered by GPU technology and distributed computing, have enabled simultaneous interaction with multiple instances of the environment (Handa et al., 2023; Gallici et al., 2024; Singla et al., 2024). This innovation allows RL agents to gather significantly more data within a given timeframe, which can result in more diverse training data and richer training signals (Horgan et al., 2018; Espeholt et al., 2018; Petrenko et al., 2020).

Parallelized data collection has the potential to facilitate better state space coverage, reduce variance in gradient estimates, and accelerate the learning process by providing a larger and more diverse data for each policy update (Schulman et al., 2017). However, the practical realization of these benefits might be hindered by intrinsic challenges within deep RL algorithms. One such challenge is the loss of plasticity, which refers to a network's ability to adapt to new information without degrading its performance on previously learned tasks (Berariu et al., 2021; Lyle et al., 2023; Juliani & Ash, 2024; Moalla et al., 2024). Recent work has also demonstrated that scaling data can often saturate an agent's performance (Singla et al., 2024).

With this work we systematically investigate the interplay between parallelized data collection, plasticity, learned representations, and sample efficiency. Our findings reveal that parallelized data collection can help mitigate common optimization challenges in deep RL, ultimately resulting in a significant positive impact on final performance. We

---

[*]Equal contribution [1]Independent Researcher [2]Mila - Québec AI Institute [3]Université de Montréal [4]Google DeepMind. Correspondence to: Johan Obando-Ceron <jobando0730@gmail.com>, Pablo Samuel Castro <psc@google.com>.

*Proceedings of the $42^{nd}$ International Conference on Machine Learning*, Vancouver, Canada. PMLR 267, 2025. Copyright 2025 by the author(s).

show that increasing the number of parallel environments ($N_{\text{envs}}$) leads to stabler training dynamics, as evidenced by reductions in weight norm and gradient kurtosis, and that this strategy is generally more effective than increasing rollout length ($N_{\text{RO}}$) for a fixed data budget (see Sec. 4 and Sec. 5). We conclude by providing guidance for maximizing data efficiency when increasing the data collected via parallelization.

## 2. Related Work

Early efforts to scale RL focused on distributing computation across multiple actors and learners. A3C (Mnih et al., 2016) introduced an asynchronous multi-threaded framework to parallelize experience collection and learning. IM-PALA (Espeholt et al., 2018) extended this idea using a decoupled actor-learner architecture with off-policy correction, enabling efficient scaling to thousands of environments. Other systems explored synchronous data collection using large batches (Stooke & Abbeel, 2018; Horgan et al., 2018), improving hardware utilization but often increasing training instability.

Recent advances in GPU-accelerated simulators such as Isaac Gym (Makoviychuk et al., 2021), EnvPool (Weng et al., 2022), and PGX (Koyamada et al., 2024) have significantly increased the throughput of data collection, enabling thousands of environment instances to run in parallel on a single device. These tools have enabled researchers to scale simulation-based learning to previously inaccessible domains, such as dexterous manipulation (Rudin et al., 2021), legged locomotion (Agarwal et al., 2022), and complex coordination tasks (Handa et al., 2023).

Several recent studies have revisited the role of parallel data collection in improving learning performance. Li et al. (2023) showed that increasing the number of environments during training can improve exploration and stabilize learning. Liu et al. (2024) demonstrated that combining parallel environment rollout with distributed learners leads to faster convergence and stronger final performance. Despite these benefits, scaling rollout length ($N_{\text{RO}}$) versus environment count ($N_{\text{envs}}$) remains underexplored.

While high-throughput simulators provide vast amounts of experience, prior work has shown that simply increasing batch size or rollout length is often suboptimal. Singla et al. (2024) showed diminishing returns when scaling data volume without accounting for structural factors such as rollout diversity or update frequency. Similar results were observed in single-environment settings, where larger batches hindered generalization and slowed adaptation (Obando Ceron et al., 2024). These findings suggest that the structure of collected data, not just its volume, is critical for effective and stable training.

Our work builds on these insights by systematically analyzing how different modes of data scaling (via $N_{\text{envs}}$ and $N_{\text{RO}}$) affect performance, sample efficiency, and network plasticity in PPO and PQN. This study isolates these variables under fixed data budgets and links their effects to optimization stability and final returns across discrete and continuous control tasks.

## 3. Background

RL is a machine learning paradigm that enables agents to map observations to actions. When these actions are executed in an environment, the environment provides a numerical reward and transitions the agent to a new state. This is typically formalized as a Markov Decision Process (MDP) (Puterman, 1994). $M := (S, A, R, P, \gamma)$, where $S$ is a finite set of states, $A$ is a finite set of actions available to the agent, $R : S \times A \to [R_{\min}, R_{\max}]$ is the reward function that gives the immediate reward received after taking action $a$ in state $s$, $P : S \times A \to \Delta(S)$ is the transition probability, where $P(s, a)(s')$ represents the probability of transitioning to state $s'$ when the agent takes action $a$ in state $s$, and $\gamma \in [0, 1)$ is a discount factor.

The primary objective of reinforcement learning is to learn a policy $\pi$ that maximizes cumulative rewards over time. A policy, which defines the behavior of an agent, is represented as $\pi : S \to \Delta(A)$. Two key approaches to obtaining the best policy are value-based (McKenzie & McDonnell, 2022) and policy-based (Sutton et al., 1999) methods. In value-based methods, the policy is determined by learning a state-action value function and selecting actions with the highest estimated value. In contrast, policy-based methods directly optimize over the policy (Schulman et al., 2017).

### 3.1. Proximal Policy Optimization (PPO)

PPO is a policy-based method that alternates between sampling and optimization (Schulman et al., 2017). In the sampling step, the algorithm collects a batch $B$ from multiple environments and rollouts. The agent then trains by sampling mini-batches from $B$ over multiple *epochs*; the size of $B$ thus determines how much data is collected, while the number of epochs determines the number of times each data point is (re-)used for learning. PPO is designed to optimize the policy while maintaining stability and preventing large, potentially destabilizing, policy updates. To achieve this, PPO introduces a clipping mechanism in the objective function, ensuring that policy updates remain within a predefined trust region:

$$\mathcal{L}^{\text{CLIP}}(\theta) = \mathbb{E}_t \left[ \min \left( r_t(\theta) A_t, \text{clip}(r_t(\theta), 1 - \epsilon, 1 + \epsilon) A_t \right) \right],$$

where $r_t(\theta) = \frac{\pi_\theta(a_t|s_t)}{\pi_{\theta_{\text{old}}}(a_t|s_t)}$ represents the probability ratio between the new policy $\pi_\theta$ and the old policy $\pi_{\theta_{\text{old}}}$. The term $A_t$ is the advantage function, which quantifies the relative

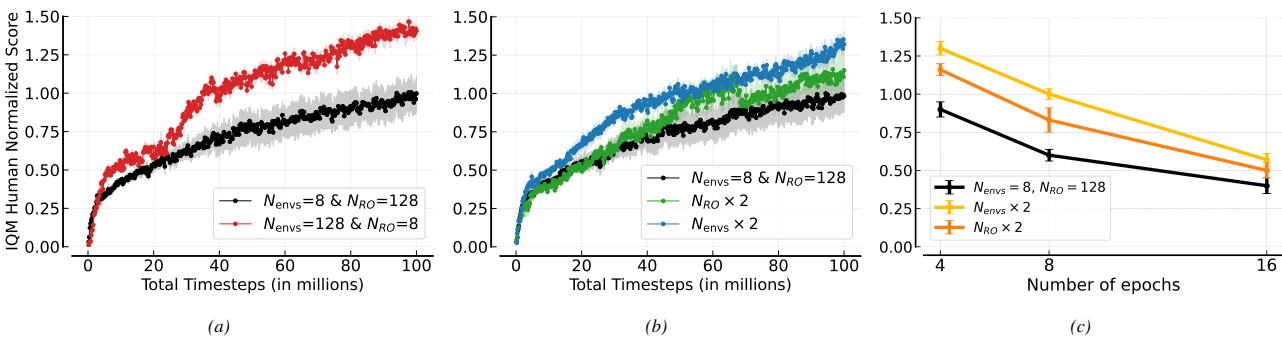

*Figure 1.* **(a)** Collecting the data with a large $N_{\text{envs}}$ is more effective; both experiments contain the same amount of data. **(b)** Scaling parallel data collection improve sample efficiency and final performance. Scaling data by increasing the $N_{\text{envs}}$ is more effective than $N_{RO}$. **(c)** Scaling parallel data collection diminishes performance degradation as the number of epochs increases. Performance collapse is better mitigated by increasing the $N_{\text{envs}}$. All the experiments were run on the Atari-10 benchmark (Aitchison et al., 2023) for PPO (Schulman et al., 2017). See Sec. 4 for training details.

improvement of an action over the expected value of the state. The hyperparameter $\epsilon$ controls the amount of clipping, ensuring that policy updates are restricted to prevent excessively large changes. The complete PPO objective is expressed as: $\mathcal{L}(\theta) = \mathcal{L}^{\text{CLIP}}(\theta) - c_1 \mathcal{L}^{\text{VF}}(\theta) + c_2 \mathcal{L}^{\text{ENT}}(\theta)$, where $\mathcal{L}^{\text{VF}}(\theta)$ represents the Value Function Loss, given by $(V_\theta(s_t) - V_t^{\text{target}})^2$, $\mathcal{L}^{\text{ENT}}(\theta)$ is the Entropy Bonus encouraging exploration, and $c_1$ and $c_2$ are coefficients that balance the contributions of each.

### 3.2. Parallelized Data Collection

The batch $B$ previously mentioned is obtained by running $N_{\text{envs}}$ environments in parallel, each for a rollout of length $N_{RO}$. Their product thus defines the *batch size* $|B| = N_{\text{envs}} \times N_{RO}$ and, as we will demonstrate below, the choice of these two factors can have a large impact on agent performance. The choice of $N_{\text{envs}}$ can have an impact on the diversity of data gathered, as governed by the support of the starting state distribution. A larger value can result in greater state-action coverage, but excessive parallelization can also result in (possibly) conflicting learning signals, highlighting the need to balance coverage and specificity. Varying $N_{RO}$, on the other hand, induces a bias-variance tradeoff: longer rollouts can result in less biased estimates of returns (approaching Monte-Carlo estimates), while excessively long rollouts may produce updates with high variance, resulting in training instabilities.

Ideally, deep RL algorithms should be maximally sample-efficient by training over $B$ multiple times. Doing so, however, has been shown to overfit to the data, leading to poor performance and plasticity (Nikishin et al., 2022; D'Oro et al., 2023; Juliani & Ash, 2024). The diversity induced by increasing $N_{\text{envs}}$ can help mitigate this risk, but it may not be sufficient when the number of epochs is high (Moalla et al., 2024).

## 4. Data collection and efficiency

Our work aims to develop a better understanding of the trade-offs mentioned above, with the aim of providing guidance for how best to scale data collection for more performant agents. In this section we examine how parallelized data collection can impact agent performance and plasticity.

### 4.1. Experimental Setup

We use the CleanRL implementation of PPO (Huang et al., 2022) with default hyperparameter settings. Our evaluation was conducted on the Arcade Learning Environment (ALE) (Bellemare et al., 2013), focusing on the Atari-10 games (Aitchison et al., 2023), which were shown to be reflective of performance on the full suite, trained for 100 million total timesteps, which is equivalent to the total number of environment steps[1]. Following the evaluation protocol proposed by Agarwal et al. (2021), each experiment was executed using 5 independent random seeds. We report the human-normalized interquartile mean (IQM) scores, aggregated across games, configurations, and seeds, along with 95% stratified bootstrap confidence intervals. In all our figures, default configurations will be shown in **black**. All experiments were performed on NVIDIA Tesla A100 GPUs, with each experiment requiring approximately 2-3 days.

### 4.2. Data collection and efficiency

**Fixed data budgets** Given a fixed data budget $B = N_{\text{envs}} \times N_{RO}$, Fig. 1(a) illustrates the impact of varying $N_{\text{envs}}$ and $N_{RO}$ while maintaining $B$ fixed. Specifically, we compare the default settings ($N_{\text{envs}} = 8$, $N_{RO} = 128$) with a variant that increases the number of environments and reduces the rollout length ($N_{\text{envs}} = 128$, $N_{RO} = 8$). This

---

[1]The number of environment interactions is fixed across all experiments when varying $N_{\text{envs}}$ or $N_{\text{RO}}$.

result suggests that, for a fixed data budget, scaling $N_{\text{envs}}$ is preferable to scaling $N_{\text{RO}}$.

**Scaling data** We next explore settings where the data budget is increased. Fig. 1(b) illustrates the impact of increasing $B$ by varying $N_{\text{envs}}$ and $N_{RO}$. While results show that increasing either leads to improved performance, increasing $N_{\text{envs}}$ appears to yield more gains, consistent with our previous result.

**Data reuse** The number of epochs controls the number of times we train over each data point. In Fig. 1(c) we explore increasing the default value of 4 by a factor of 2 and 4, while simultaneously increasing $N_{\text{envs}}$ and $N_{\text{RO}}$. Consistent with prior works (Nikishin et al., 2022; Sokar et al., 2023), overly training on the same data, via increasing the number of epochs, negatively impacts performance. However, it can be seen that the gains obtained from increasing data, in particular by scaling $N_{\text{envs}}$, can help to counteract this degradation. For instance, the performance decline when training twice as many times over the data (with 8 epochs) is completely overcome by doubling the number of environments.

**Data diversity** In on-policy RL, the diversity of collected trajectories plays a key role in determining agent performance by influencing the quality of the training signal. Fig. 2 shows how varying the number of parallel environments ($N_{\text{envs}}$) and rollouts per environment ($N_{\text{RO}}$) affects the coverage of the learned state distribution. Increasing $N_{\text{envs}}$ enhances spatial diversity by exposing the policy to a wider range of initial conditions and environment stochasticity, while increasing $N_{\text{RO}}$ improves temporal depth but may induce stronger trajectory correlations. Configurations with higher $N_{\text{envs}}$ (right column) consistently yield greater state-space coverage and higher performance across Atari games, as measured by a visitation-based metric. This highlights that increasing the $N_{\text{envs}}$, while keeping total samples constant, can lead to diverse policy updates. Fig. 9 further supports this observation by showing consistent improvements in state-space coverage across different $N_{\text{envs}}$ values.

While longer rollouts might improve temporal credit assignment, our results suggest the gains from better state-action coverage induced by more environments are more effective at improving performance and mitigating overfitting. It is likely that this tradeoff is not fixed throughout training, and future work could explore adjusting $N_{\text{envs}}$ and $N_{\text{RO}}$ dynamically throughout training. Taken together, our results in this section can be summarized as follows.

> When dealing with a fixed or scaling data budget, or when increasing the number of epochs, it is more effective to scale $N_{\text{envs}}$ over $N_{\text{RO}}$.

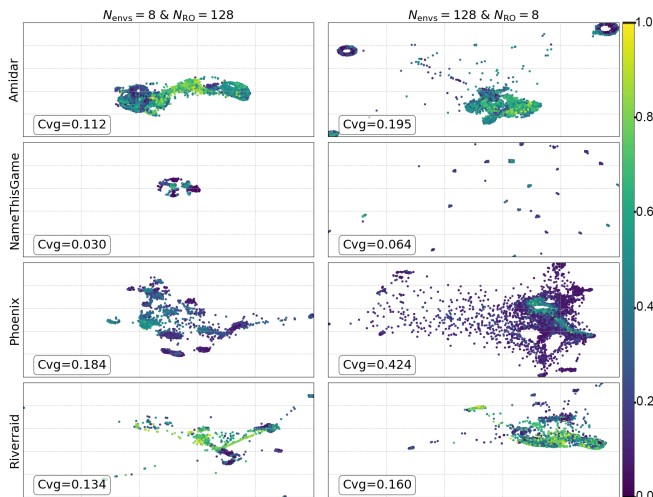

*Figure 2.* **State-space coverage under varying parallel rollout configurations across Atari games.** (Left) fewer environments, more rollouts per environment ($N_{\text{envs}} = 8$, $N_{\text{RO}} = 128$). (Right) more environments, fewer rollouts ($N_{\text{envs}} = 128$, $N_{\text{RO}} = 8$). Increasing the number of environments improves spatial coverage and distributional spread, as quantified by the coverage metric (Cvg) shown in each plot. See Sec. B.7 for Cvg metric details. The color scale indicates the critic value associated with each projected point: higher critic values are shown in yellow, and lower values in blue.

## 5. Analyses

Having demonstrated the benefits of scaling environments, in this section we explore its connection to various learning dynamics and algorithmic components, with the aim to better understand the reasons for the observed gains.

### 5.1. Learning dynamics

To assess how learning dynamics are affected by changing data collection strategies, in Fig. 3 we examine the following metrics throughout training in four representative games: *feature rank, dormant neurons, weight norm* and *gradient kurtosis*; these metrics can serve as proxy indicators of feature collapse and loss of plasticity. The results on the remaining games are provided in Fig. 10, and results on extra games in Fig. 15.

**Feature Rank** Feature rank reflects the effective dimensionality of the representations learned by the agent's neural network (see Sec. B.1 for further details). Higher feature rank suggests richer and more diverse representations, which can contribute to improved policy learning. We consistently observe an increase in feature rank when increasing $N_{\text{envs}}$, suggesting that the resulting increased data diversity can improve learned representations. This is consistent with prior techniques, such as data augmentation, which have been shown to aid in learned representations.

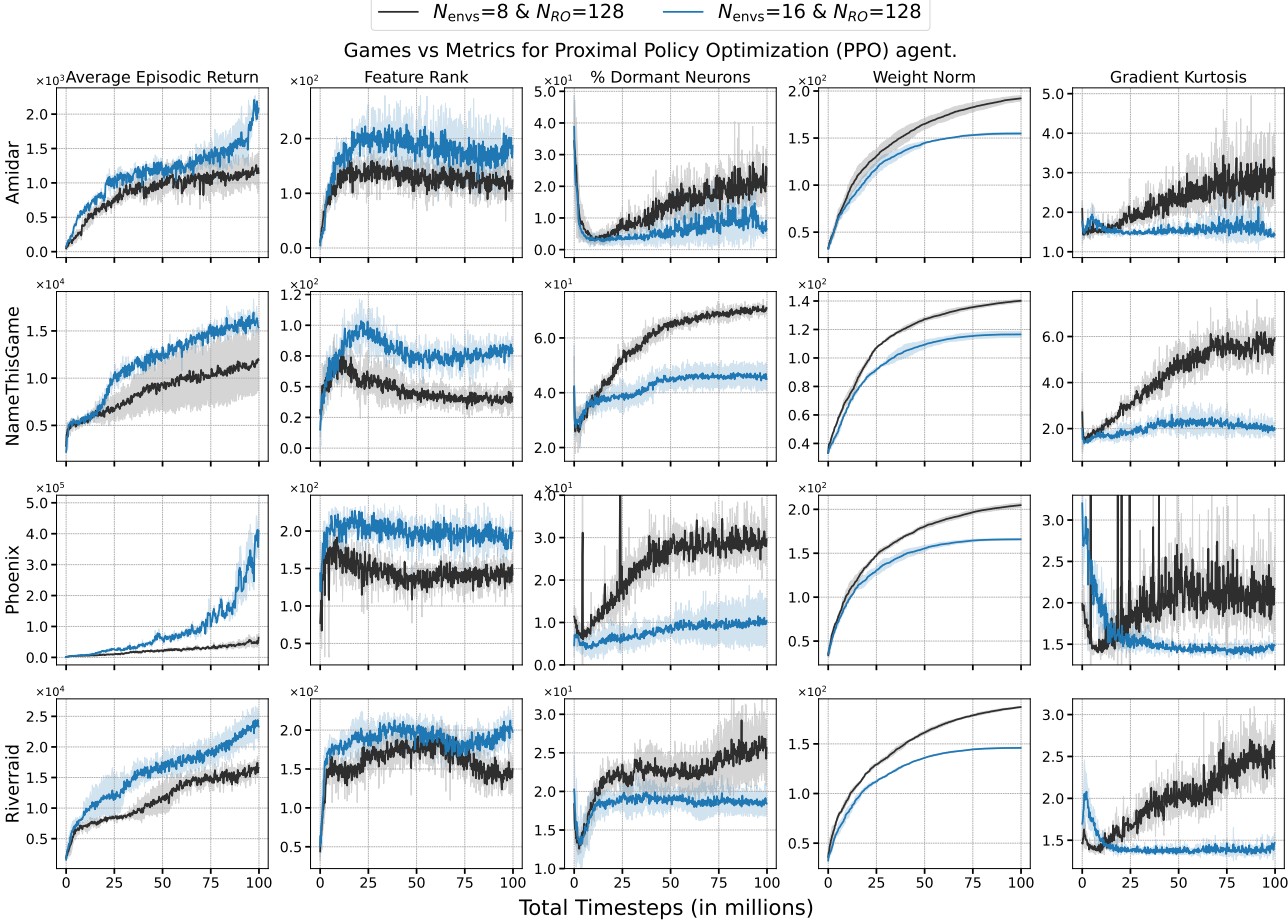

*Figure 3.* **Empirical analyses for four representative games with different amount of parallel data for PPO (Schulman et al., 2017).** From left to right: training returns, feature rank, dormant neurons percentage (Sokar et al., 2023), weight norm and gradient kurtosis. All results averaged over 5 seeds, shaded areas represent 95% confidence intervals. See Sec. 4 for training details.

**Dormant Neurons** We measure the fraction of neurons that have gone dormant, or inactive, during training (see Sec. B.2 for details), which can indicate inefficient usage of network capacity (Sokar et al., 2023; Ceron et al., 2024c;b). We observe a negative correlation between $N_{\text{envs}}$ and the level of neuron dormancy. This suggests that the increased data diversity induced by more environments can help mitigate dormancy and thereby increase network utilization and plasticity.

**Weight Norm** Weight norm measures the magnitude of the network weights (see Sec. B.3 for details) and has been shown to be correlated with training instability (Krogh & Hertz, 1991; Bartlett, 1996; Hinton et al., 2012; Neyshabur et al., 2015). Our results show an inverse correlation between $N_{\text{envs}}$ and weight norm, suggesting that increased data collection via more parallelization can help stabilize network training.

**Gradient Kurtosis** Gradient kurtosis (see Sec. B.4 for details) quantifies the sharpness or variability of gradient distri-

butions (Garg et al., 2021). High kurtosis can indicate sharp minima and unstable training which implies an increase in heavy-tailedness of the gradient distribution, whereas lower kurtosis may correspond to smoother gradients. We observe a strong negative correlation between gradient kurtosis and $N_{\text{envs}}$; it is possible that the increased state-action diversity induced by a higher number of parallel environments has a regularizing effect on the gradients, which in turn leads to stable learning.

**Policy Variance and ESS** Policy variance measures the variability in the $\pi(\cdot)$ and is correlated with the network's "churn" (Schaul et al., 2022). Effective Sample Size (ESS) measures the proportion of independent and high-quality samples contributing to policy updates, with higher ESS values suggesting more effective learning (Martino et al., 2017). Fig. 4 shows that increasing $N_{\text{envs}}$ leads to lower policy variance and higher ESS, which further helps explain the observed increased performance.

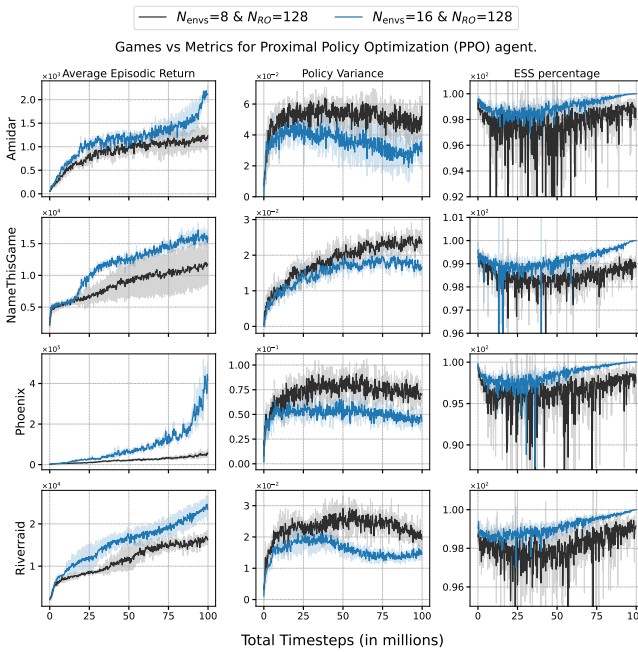

*Figure 4.* **Increasing $N_{\text{envs}}$ for parallel data collection leads to lower policy variance and higher ESS,** which results in a higher average episodic return. All results averaged over 5 seeds, shaded areas represent 95% confidence intervals. See Sec. 4 for training details.

---

**Key observations on increased $N_{\text{envs}}$:**

- Increases representational diversity and prevents feature collapse.

- Leads to broader state space coverage, ensuring better exploration.

- Results in networks that are both more expressive and exhibit greater plasticity.

- Reduces weight norm, which tend to result in stabler learning.

- Exhibits low heavy-tailedness in gradients, which offers optimization stability.

- Reduces policy variance and increases ESS, leading to stabler and more effective learning.

---

### 5.2. Decoupled architectures

The architectural choice between shared and separate encoders for the actor and critic networks can significantly influence an agent's performance (Cobbe et al., 2021). While shared encoders streamline the model and reduce computational overhead, they may inadvertently constrain learning capacity by requiring a unified representation to simultaneously optimize both policy and value functions. In contrast, using dedicated encoders allows each network to indepen-

dently optimize its feature representations, potentially enhancing learning efficiency and performance, particularly in complex environments. This separation can be especially beneficial when scaling parallel data, given that in Sec. 5.1 we demonstrated that scaling $N_{\text{envs}}$ helps in learning more expressive and robust representations.

Fig. 5 shows that when employing separate networks for the actor and critic, agents benefit more from scaling parallel data collection, leading to improved final performance. However, our findings suggest that separate encoders alone do not always provide substantial performance gains over shared architectures unless paired with large-scale data collection. This indicates that data diversity plays a crucial role in maximizing the benefits of independent actor-critic representations.

### 5.3. Hyper-parameter sensitivity

The effect of varying one algorithmic parameter is known to be tied to the settings of other hyper-parameters. Given our focus on increasing batch size, often via increasing rollouts, in this section we investigate the sensitivity of our findings to the choice of two related components: the learning rate and the discount factor $\gamma$.

**Learning rate** The learning rate plays a pivotal role in the stability of RL optimization (Andrychowicz et al.; Ceron et al., 2024a), and is generally advised to be re-tuned for larger batch sizes (Shallue et al., 2019). It bears questioning whether the strong performance gains we have thus far observed by increasing $N_{\text{envs}}$ is sensitive to the particular choice of learning rate. Indeed, Hilton et al. (2022) suggest scaling the learning rate by a factor proportional to the change in $B$; specifically, either $k$ or $\sqrt{k}$, where $k$ is the multplicative factor used to increase $N_{\text{envs}}$. Fig. 11 illustrates that the current default learning rate generally yields the best performance. While we observe some variability with different choices for the learning rate, these differences are not significant.

**Length of Rollouts and $\gamma$** The choice of $N_{RO}$ can significantly impact training efficiency and stability (Ceron et al., 2024a). A larger $N_{RO}$ provides more temporally extended trajectories, allowing reduced estimation bias by incorporating more future rewards. However, high values of $N_{RO}$ can result in over-correlated data with higher variance, negatively affecting updates. As Fig. 1 demonstrates, we can scale data collection via increasing $N_{\text{envs}}$ and reducing $N_{RO}$.

The choice of the discount factor $\gamma$ is intimately tied with the rollout length, as it affects how much transitions are weighted along a trajectory, as well as in general advantage estimation (Schulman et al., 2015), which is a core component of PPO. Thus, it is worth investigating how sensitive

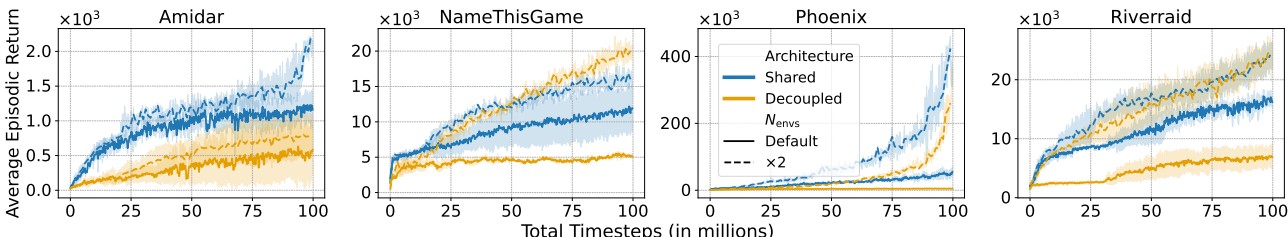

*Figure 5.* **When using separate networks to represent the policy and value function (Cobbe et al., 2021), scaling parallel data collection improves final performance.** Using decoupled architectures with default settings collapses performance; this collapse is mitigated by scaling $N_{envs}$. See Sec. 4 for training details.

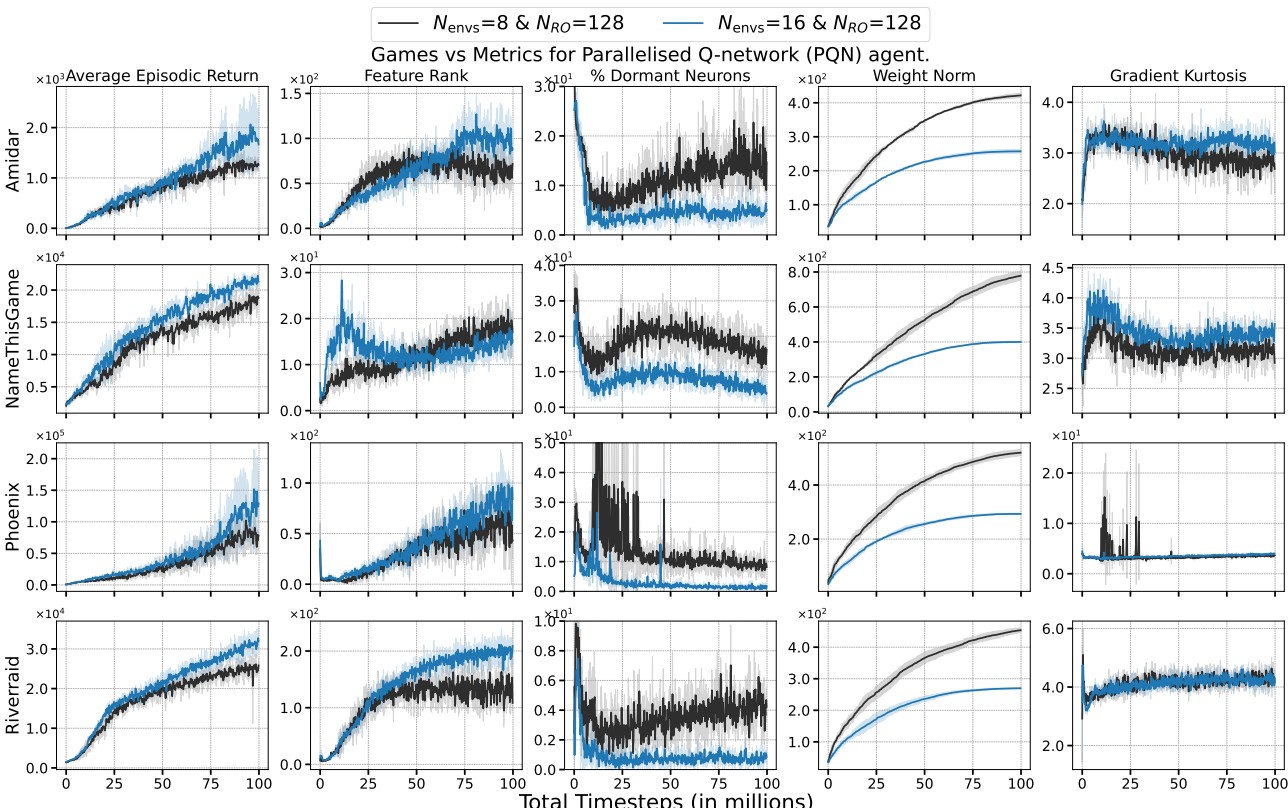

*Figure 6.* **Scaling parallel data collection on Parallelised Q-network (PQN) (Gallici et al., 2024) marginally mitigates representation deterioration and slightly improves final performance.** From left to right: training returns, feature rank, dormant neurons percentage (Sokar et al., 2023), weight norm and gradient kurtosis. All results averaged over 5 seeds, shaded areas represent 95% confidence intervals. See Sec. 4 for training details.

the performance gains observed in Fig. 1(b) are to different values of $\gamma$. In Fig. 12 we swept over a number of values for $\gamma$ and found that the observed performance gains remain unaffected.

# 6. Analysis in other settings

We extend our analysis to value-based methods, which offer alternative learning paradigms, and assess the impact of parallel data collection across additional environments to evaluate the generalization of our findings.

## 6.1. Value-based methods

PPO (Schulman et al., 2017) and Parallel Q Network (PQN) (Gallici et al., 2024)[2] both collect on policy parallel data, but they differ fundamentally in the nature of their loss functions. PQN is a value-based learning method that simplifies and improves deep temporal difference learning approaches such as DQN (Mnih et al., 2015). It eliminates the need for both a replay buffer and a target network. PQN collects a batch $B$ of experiences from multiple environments and rollouts and samples mini-batches for training, similar to PPO.

---

[2]We use the PQN implementation from CleanRL.

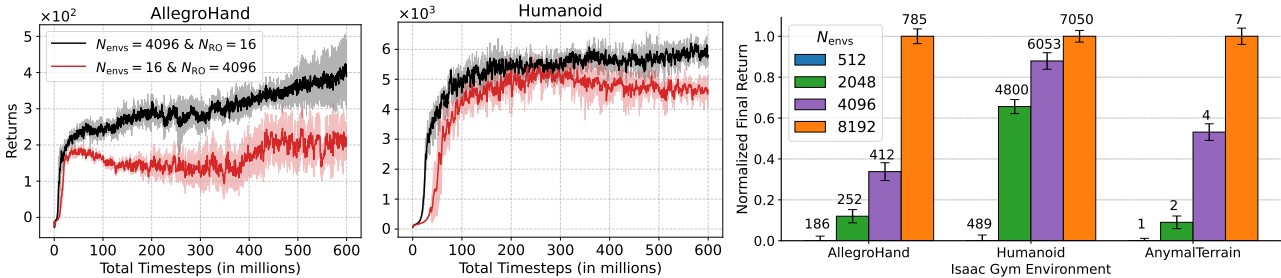

*Figure 7.* **Proximal Policy Optimization (PPO)** (Schulman et al., 2017; Huang et al., 2022) on Isaac Gym environments (Makoviychuk et al.). (Left) Scaling data by increasing the $N_{\text{envs}}$ is more effective than $N_{RO}$; both experiments contains the same amount of data. (Right) Increasing $N_{\text{envs}}$ for parallel data collection improves final performance. We report returns over 5 runs for each experiment.

The learning objective for PQN is to minimize the mean squared error (MSE) between the predicted Q-values $Q_\phi(s_i, a_i)$, where $\phi$ are the network parameters, and the Bellman target for each sample $i$ from $B$:

$$\mathcal{L}(\phi) = \frac{1}{B} \sum_{i=1}^{B} \left( \text{target}_i - Q_\phi(s_i, a_i) \right)^2,$$

$\text{target}_i = r_i + \gamma \cdot \max_{a'} Q_\phi(s'_i, a')$; where $s_i$, $a_i$, $r_i$, $a'_i$ and $s'_i$ is sampled from the current batch. We conduct a similar analysis as with PPO to examine if the same benefits can be observed from scaling $N_{\text{envs}}$ and $N_{RO}$. In Fig. 6 and Fig. 14 we can see that, although larger values of $N_{\text{envs}}$ do not appear to yield significant performance improvements, we do observe some mild improvements on some of the learning dynamics metrics. We observe a similar trend across the remaining Atari-10 suite; see Fig. 13. Although there are a number of differences between PPO and PQN, we hypothesize that these qualitative differences are largely due to the choice of loss function and, more generally, the distinction between value-based and policy-based methods.

### 6.2. Evaluating on a separate benchmark

To further assess the generalization and scalability of parallel data collection, we extend our analysis to two environments from the Procgen suite (Cobbe et al., 2020), which uses procedural generation for each level; and three Isaac Gym environments (Makoviychuk et al.). In Fig. 7 and Fig. 8 we observe the same general tendency: increasing $B$ improves performance, but it is more effective to do so by scaling $N_{\text{envs}}$.

### 6.3. Parallel environments and exploration

In sparse-reward or hard-exploration settings, increasing the number of parallel environments enhances the chance of encountering rare but informative events by amplifying the agent's exposure to diverse trajectories. Unlike increasing $N_{\text{RO}}$, which can yield temporally correlated data, scaling the $N_{\text{envs}}$ provides independent instantiations of environment

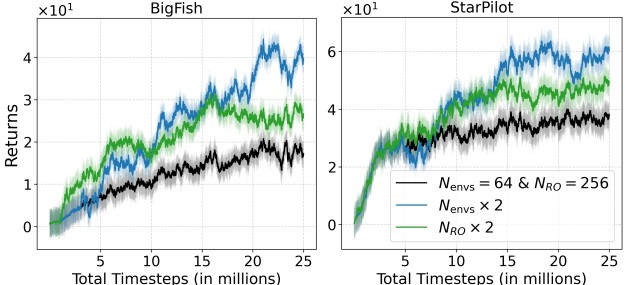

*Figure 8.* **Scaling parallel data collection improves final performance on the BigFish and StarPilot games from the Procgen Benchmark** (Cobbe et al., 2020). We observe that both increasing $N_{\text{envs}}$ and $N_{RO}$ leads to greater performance, with $N_{\text{envs}}$ the more effective of the two, as discussed in Sec. 3.2.

stochasticity and initial states, reducing redundancy and promoting wider coverage of the state space. As shown in Fig. 9, higher $N_{\text{envs}}$ configurations lead to broader empirical support in the embedded state distribution, suggesting improved exploration dynamics even under fixed sample budgets. This structural diversity can complement algorithmic exploration strategies, leading to more robust and sample efficient learning. This effect is particularly beneficial in challenging exploration tasks such as MONTEZUMA'S RE-VENGE and PHOENIX, where progress depends on rare event chains. Fig. 15 and Fig. 16 show mid-training improvements in these environments, motivating further investigation.

## 7. Discussion

The findings of this study provide insights into the fundamental trade-offs in RL training with regards to data collection using parallel actors (Singla et al., 2024). Our results indicate that larger dataset sizes, driven by increased $N_{\text{envs}}$ or $N_{\text{RO}}$, can enhance final agent performance. This aligns with previous research emphasizing the importance of data availability in deep learning and RL for improving generalization and robustness (Taiga et al., 2023; Kumar et al., 2022). Our analysis suggests that scaling $N_{\text{envs}}$ is a more effective strategy than increasing $N_{\text{RO}}$, which can be an im-

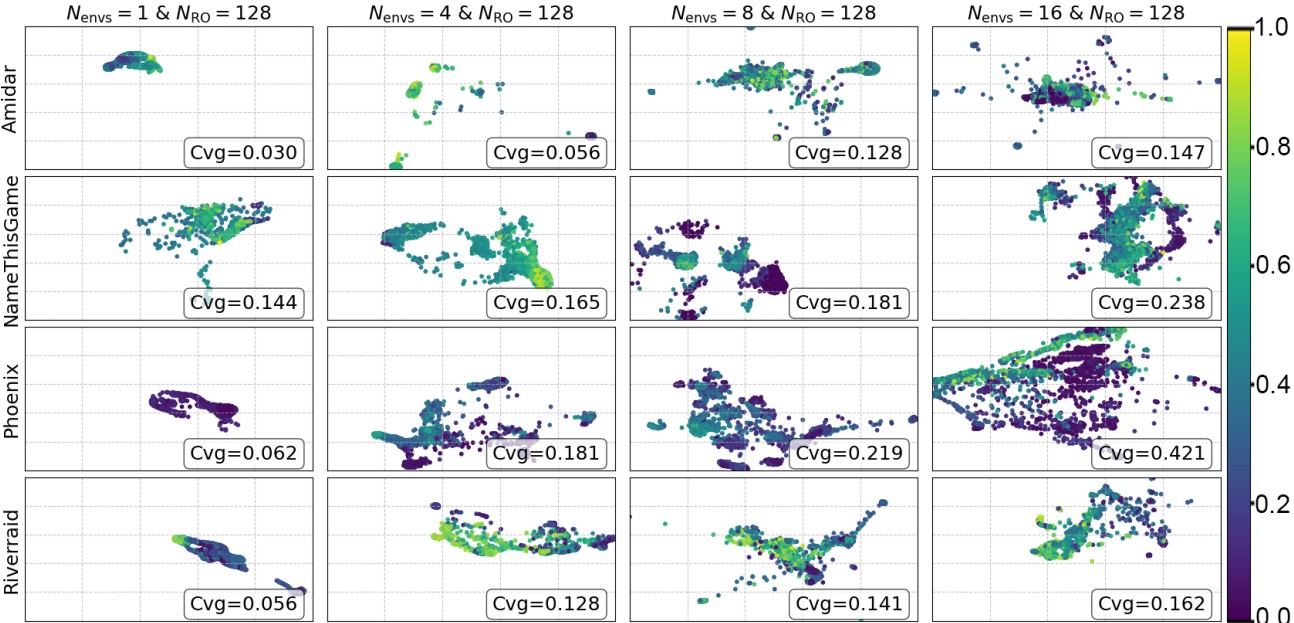

*Figure 9.* **State-space coverage under varying levels of parallel data collection.** Increasing the $N_{\text{envs}}$ leads to improved spatial coverage across Atari games. This suggests that scaling $N_{\text{envs}}$ promotes broader exploration and diverse training data under a fixed sample budget. The color scale indicates the critic value associated with each projected point: higher critic values are shown in yellow, and lower values in blue.

portant consideration in settings where one is constrained by computational resources.

Recent works (Moalla et al., 2024; Juliani & Ash, 2024) have observed representation deterioration and loss of plasticity in the on-policy deep RL setting, particularly on PPO. Juliani & Ash (2024) show that plasticity loss is widespread under domain shift in this regime and that several existing methods designed to address it in other contexts (Sokar et al., 2023; Nikishin et al., 2022) often fail, sometimes even performing worse than taking no intervention at all. We provide evidence of the impact of parallel data collection in addressing optimization challenges, such as feature collapse and loss of plasticity.

In this work, we use GPU-vectorized PPO and PQN algorithms to explore how large-batch data collection affects network plasticity and learning dynamics. We find that scaling the batch of data can mitigate representation collapse and reduce dormant neurons. Additionally, we observe a negative correlation between the number of environments $N_{\text{envs}}$ and both weight norm and gradient kurtosis, suggesting that increasing data collection through parallelization can stabilize network training.

Scaling the batch of data via increased parallel environments can be enhanced by exploring alternate network architectures. In Fig. 5 we demonstrated that, by doubling $N_{\text{envs}}$, PPO is able to avoid collapse and obtain strong performance. The relatively low performance gains observed when scaling the batch of data in PQN warrants a more detailed analysis,

given the similarity to PPO in terms of data collection. We hypothesize that this is due to their difference in loss functions; if so, investigating this further could shed light on the fundamental and practical differences between policy- and value-based methods.

**Future work:** In our analyses, like in most prior works, we have maintained the values of $N_{\text{envs}}$ and $N_{\text{RO}}$ fixed throughout training. This is most likely sub-optimal as it is known that the learning dynamics in RL vary throughout training (Nikishin et al., 2022; Lyle et al., 2024) Thus, even when maintaining the data budget fixed, the optimal trade-off between $N_{\text{envs}}$ and $N_{\text{RO}}$ will likely vary throughout training. Future work will explore setting these values, and the size of $B$ itself, dynamically. The recent growth in popularity and impact of machine learning, in particular large language models (LLMs), has been largely driven by the ability of supervised learning methods to leverage massive amounts of *pre-existing* data. Online RL algorithms typically do not have access to this type of dataset, and must instead collect data throughout training. While parallelization can help speed this collection, agent performance is still hampered by many of the standard challenges in RL, such as those outlined in the deadly triad (Sutton & Barto, 2018). If RL is to achieve scale and impact comparable to that of LLMs (beyond its use for fine-tuning LLMs), we need to develop a better understanding of the challenges, and successful strategies, in parallel data collection. Our work is a step in that direction, and provides guidance for future research to continue developing these insights.

## Acknowledgements

The authors would like to thank Ghada Sokar, Roger Creus Castanyer, Olya Mastikhina, Dhruv Sreenivas, Ali Saheb Pasand, Ayoub Echchahed and Gandharv Patil for valuable discussions during the preparation of this work. Ghada Sokar deserves a special mention for providing us valuable feed-back on an early draft of the paper. We thank the anonymous reviewers for their valuable help in improving our manuscript.

We want to acknowledge funding support from Google, CIFAR AI and compute support from Digital Research Alliance of Canada and Mila IDT. We would also like to thank the Python community (Van Rossum & Drake Jr, 1995; Oliphant, 2007) for developing tools that enabled this work, including NumPy (Harris et al., 2020), Matplotlib (Hunter, 2007), Jupyter (Kluyver et al., 2016), and Pandas (McKinney, 2013).

## Impact Statement

This paper presents work whose goal is to advance the field of Machine Learning. There are many potential societal consequences of our work, none which we feel must be specifically highlighted here.

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

# A. Training Details

In this section, we offer a detailed overview of the hyperparameters and configurations for each experimental section. To ensure robustness and reliability of our results, each experimental setup is conducted with five random seeds.

**Tasks**   We evaluate PPO and PQN on 10 games from the Arcade Learning Environment (Bellemare et al., 2013). We use the same set of games evaluated by Aitchison et al. (2023). This includes: Amidar, Bowling, Frostbite, KungFuMaster, Riverraid, BattleZone, DoubleDunk, NameThisGame, Phoenix, Qbert.

**Hyper-parameters**   We use the default hyper-parameter values for PPO (Schulman et al., 2017) and PQN (Gallici et al., 2024) agents. We share the details of these values in Tab. 1 and Tab. 2.

*Table 1.* Default hyper-parameters setting for PPO and PQN agents.

| | Atari | |
| --- | --- | --- |
| Hyper-parameter | PPO | PQN |
| Adam's ($\epsilon$) | 1e-5 | 1e-8 |
| Adam's learning rate | 2.5e-4 | 2.5e-4 |
| Conv. Activation Function | ReLU | ReLU |
| Convolutional Width | 32,64,64 | 32,64,64 |
| Dense Activation Function | ReLU | ReLU |
| Dense Width | 512 | 512 |
| Normalization | None | LayerNorm |
| Discount Factor | 0.99 | 0.99 |
| Exploration $\epsilon$ | 0.01 | 0.011 |
| Exploration $\epsilon$ decay | 250000 | 250000 |
| Number of Convolutional Layers | 3 | 3 |
| Number of Dense Layers | 2 | 2 |
| Reward Clipping | True | True |
| Weight Decay | 0 | 0 |

*Table 2.* Default hyper-parameters setting for PPO agent.

| | Procgen |
| --- | --- |
| Hyper-parameter | PPO |
| Adam's ($\epsilon$) | 1e-5 |
| Adam's learning rate | 5e-4 |
| Conv. Activation Function | ReLU |
| Convolutional Width | 16,32,32 |
| Dense Activation Function | ReLU |
| Dense Width | 256 |
| Normalization | None |
| Discount Factor | 0.999 |
| Exploration $\epsilon$ | 0.01 |
| Exploration $\epsilon$ decay | 250000 |
| Number of Convolutional Layers | 3 |
| Number of Dense Layers | 2 |
| Reward Clipping | True |
| Weight Decay | 0 |

# B. Metrics

### B.1. Feature Rank

This metric evaluates representation quality in deep RL by finding the smallest subspace retaining 99% variance, improving interpretability, efficiency, and stability. A high feature rank indicates diverse representations and it is computed using the approximate rank from Yang et al., 2019; Moalla et al., 2024.

$$\sum_{i=1}^{k} \frac{\sigma_i^2}{\sum_{j=1}^{n} \sigma_j^2} \geq \tau$$

Where $\sigma_i$ denotes the singular values of the feature matrix, $n$ represents the total number of singular values, and $\tau$ is a threshold (e.g., 99%) used to determine the rank. The value $k$, referred to as the "feature rank," corresponds to the smallest number of principal components (or singular values) needed to preserve at least $\tau$ of the total variance in the data.

### B.2. Dormant Neuron

This metric quantifies the number of inactive neurons with near-zero activations, limiting network expressivity. It helps detect learning inefficiencies and improve model performance. A high number of dormant neurons means many are inactive or rarely contribute to the model's output. Its computation follows Sokar et al., 2023.

$$\frac{\sum_{i=1}^{N} \mathbf{1}(|a_i| < \epsilon)}{N} \times 100,$$

where $N$ is the total number of neurons, $a_i$ is the activation of neuron $i$, $\epsilon$ is a small threshold (e.g., $10^{-5}$), and $\mathbf{1}$ is the indicator function.

### B.3. Weight Norm

This metric quantifies the magnitude of neural network weights, helping evaluate model complexity, stability, generalization, and overfitting risks. A high weight norm indicates that the model's parameters have large magnitudes, reducing its ability to fit new targets over time. It is calculated as in Moalla et al., 2024; Lyle et al., 2023.

$$\|\theta\|_2 = \sqrt{\sum_i \theta_i^2}$$

where $\theta_i$ are the weights of the layer.

### B.4. Kurtosis

This metric evaluates the sharpness and tail heaviness of weight gradients, highlighting distribution shape and outliers. High kurtosis indicates extreme gradient values, heavy-tailed distribution, and large gradient magnitude disparities. Following Garg et al., 2021, we additionally apply a logarithmic transformation to minimize the impact of extreme values and emphasize differences between small and large gradients.

$$K = \frac{\mathbb{E}[(L - \mu_L)^4]}{\sigma_L^4}$$

where $L_i = \log(|G_i| + \epsilon)$ are the log-transformed absolute gradients, $\mu_L$ is the mean of the log-transformed gradients, $\sigma_L$ is the variance of the log-transformed gradients, $G_i$ represents each individual gradient and $\epsilon$ is a small positive constant to prevent undefined values when $G_i = 0$.

### B.5. Effective Sample Size

This metric estimates the number of independent samples in importance sampling by assessing weight dispersion, reflecting sampling efficiency and estimator accuracy. A high ESS indicates low weight variance, efficient sampling, and accurate estimator performance. It is computed using the ESS approximation from Martino et al. 2017.

$$ESS = \frac{1}{\sum_{i=1}^{N} \tilde{r}_i^2(\theta)}$$

where $\tilde{r}_i(\theta)$ represents the normalized ratio, which is computed as:

$$\tilde{r}_i(\theta) = \frac{r_i(\theta)}{\sum_{j=1}^{N} r_j(\theta)}$$

where $r_i(\theta) = \frac{\pi_\theta(a_t|s_t)}{\pi_{\theta_{\text{old}}}(a_t|s_t)}$ represents the probability ratio between the new policy $\pi_\theta$ and the old policy $\pi_{\theta_{\text{old}}}$. The variable $N$ denotes the number of samples drawn from $\pi_{\theta_{\text{old}}}$ and weighted according to $r_i(\theta)$. The ESS takes values in the range $1 \leq ESS \leq N$. Based on this, we propose using the percentage of the ESS as:

$$ESS\% = \frac{\text{ESS}}{N} \times 100$$

## B.6. Policy variance

The metric quantifies action probability dispersion across states, reflecting policy diversity and consistency within a batch. It is computed similarly to the policy variance measure in Moalla et al. 2024.

$$\sigma_{\text{policy}}^2 = \frac{1}{A} \sum_{a=1}^{A} \sigma_a^2$$

where $A$ represents the total number of possible actions in the action space of the policy and $\sigma_a^2$ is the variance of the probability of action $a$ across different states, computed as:

$$\sigma_a^2 = \frac{1}{B} \sum_{i=1}^{B} (p_{i,a} - \bar{p}_a)^2$$

where $B$ is the number of states (batch size), $p_{i,a}$ is the probability of selecting action $a$ in state $s_i$ and $\bar{p}_a$ is the mean probability of selecting action $a$ across all states.

## B.7. UMAP and Coverage Metric (Cvg)

This visual and numerical metric allows us to quantify the cumulative dispersion of batch data across iterations. We begin by projecting the high-dimensional batch data into a 2D space using UMAP (McInnes et al., 2018), a popular technique for visualizing high-dimensional data. UMAP is particularly suitable for this task as it preserves both the local neighborhood structure and the global layout of the data manifold, enabling a faithful representation of how the internal representations are distributed over time.

Let the resulting 2D projection be denoted as:

$$\mathcal{D} = \{(x_i, y_i) \mid i = 1, \ldots, N\}, \quad \text{where } x_i, y_i \in [0, 1)$$

We partition the 2D space into a uniform grid of size $G \times G$. Each point $(x_i, y_i)$ is assigned to a grid cell as follows:

$$\text{idx}_i^x = \min\left(\lfloor x_i \cdot G \rfloor, G-1\right), \quad \text{idx}_i^y = \min\left(\lfloor y_i \cdot G \rfloor, G-1\right)$$

We define the set of occupied grid cells as:

$$\mathcal{O} = \{(\text{idx}_i^x, \text{idx}_i^y) \mid i = 1, \ldots, N\}$$

Finally, the spatial coverage is computed as the fraction of grid cells visited by at least one point:

$$\text{Coverage} = \frac{|\mathcal{O}|}{G^2}$$

where $|\mathcal{O}|$ denotes the number of unique occupied cells, and $G^2$ is the total number of cells in the grid.

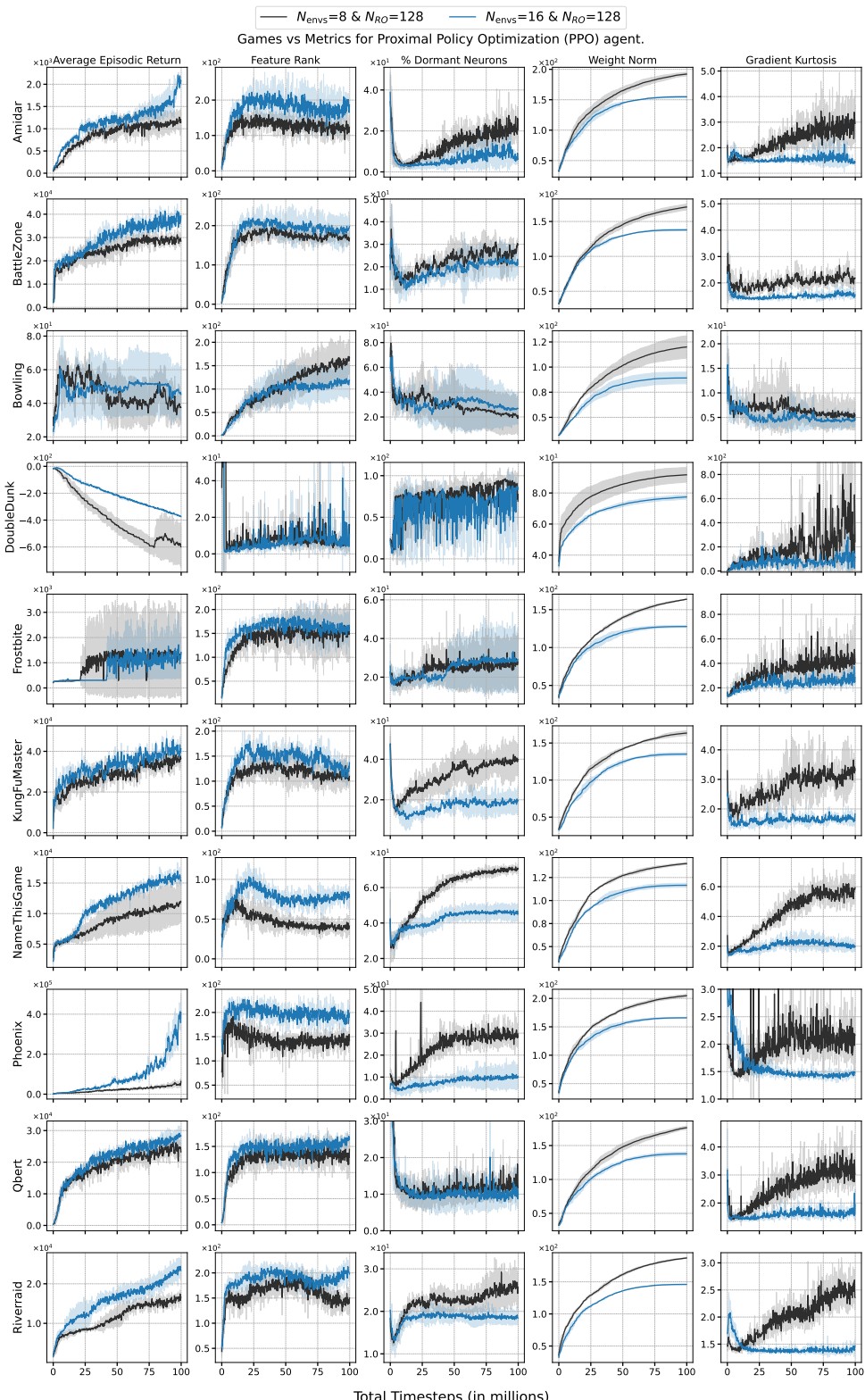

*Figure 10.* **Empirical analyses for ALE games with different amount of parallel data for PPO (Schulman et al., 2017).** From left to right: training returns, feature rank, dormant neurons percentage (Sokar et al., 2023), weight norm and gradient kurtosis. All results averaged over 5 seeds, shaded areas represent 95% confidence intervals. See Section 4.1 for training details.

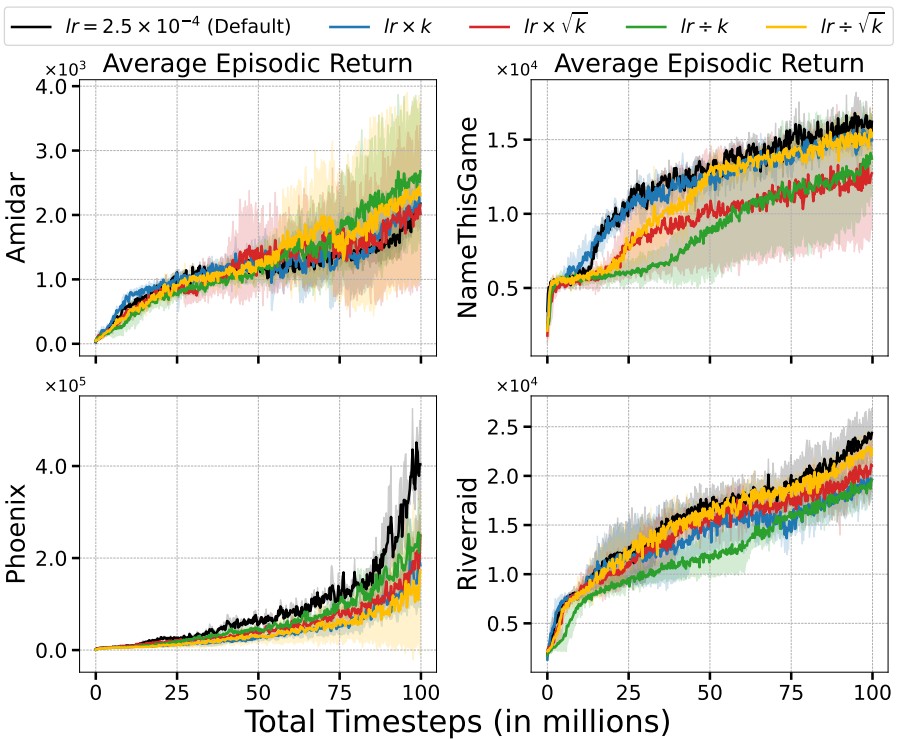

*Figure 11.* **Varying the learning rate by a factor proportional to the increase to** $N_{\textbf{envs}}$. In this figure $N_{\text{envs}}$ was scaled by a factor of 2, as Hilton et al. (2022) suggest increasing the learning accordingly. In addition to that increase, we explore decreasing the learning by the same factor. Consistent with Hilton et al. (2022). PPO is robust to learning rate changes; the default learning is $2.5 \times 10^{-4}$.

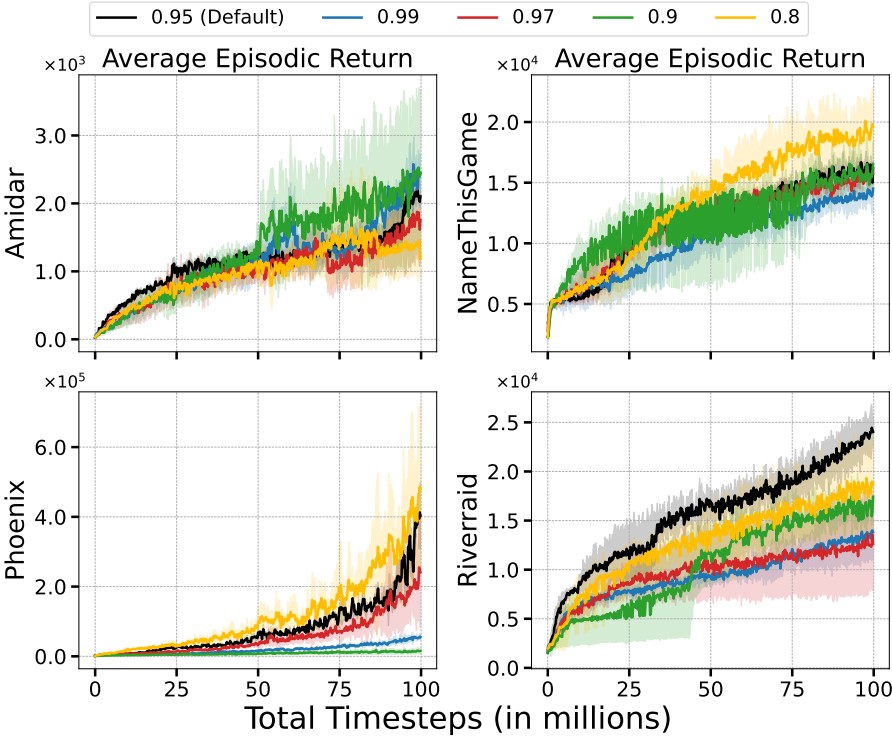

*Figure 12.* **Performance gains from** $N_{\textbf{RO}} \times 2$ **are unaffected by varying** $\gamma$, where the default value is $\gamma = 0.95$.

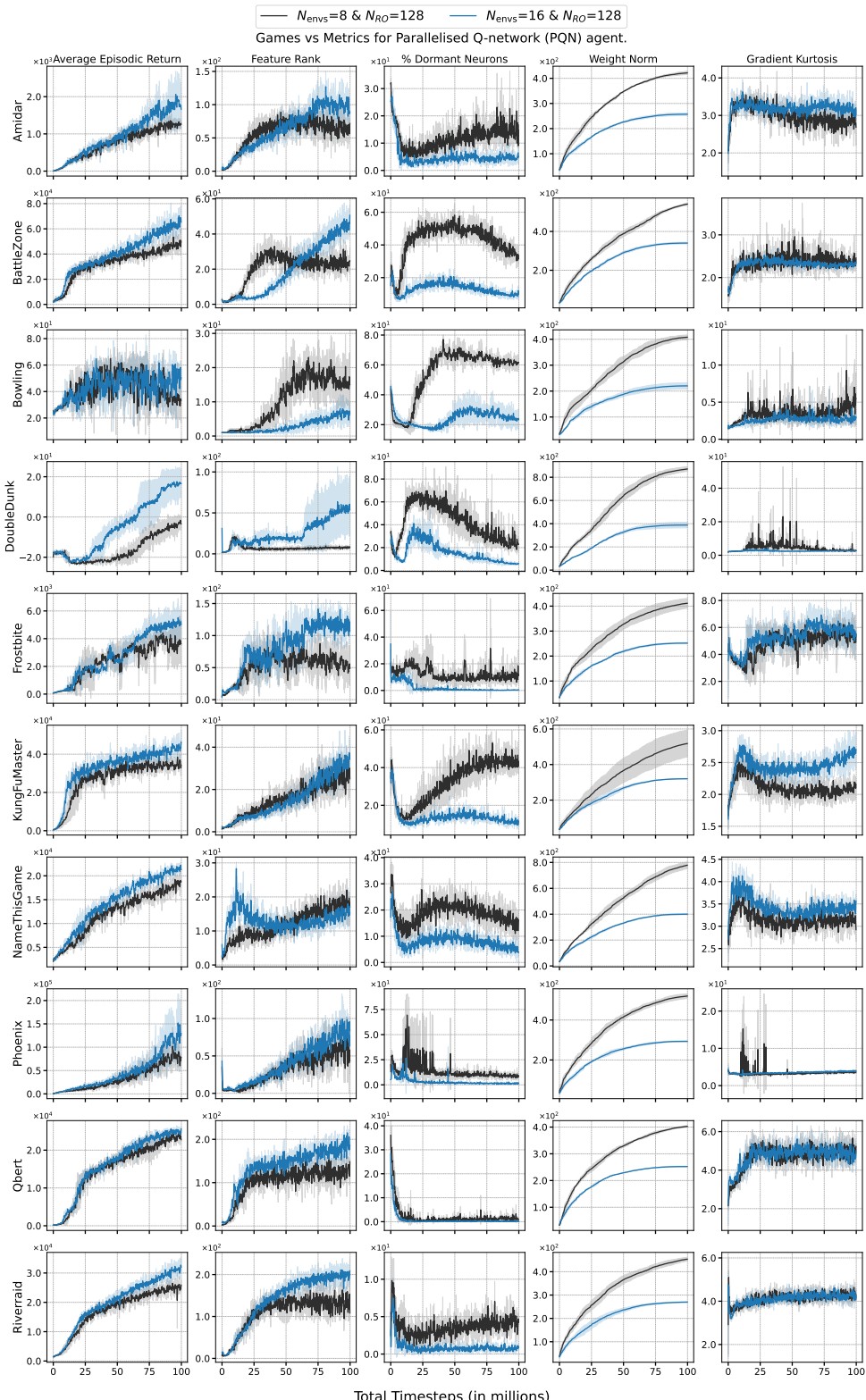

*Figure 13.* **Empirical analyses for ALE games with different amount of parallel data for PQN (Gallici et al., 2024).** From left to right: training returns, feature rank, dormant neurons percentage (Sokar et al., 2023), weight norm and gradient kurtosis. All results averaged over 5 seeds, shaded areas represent 95% confidence intervals. See Section 4.1 for training details.

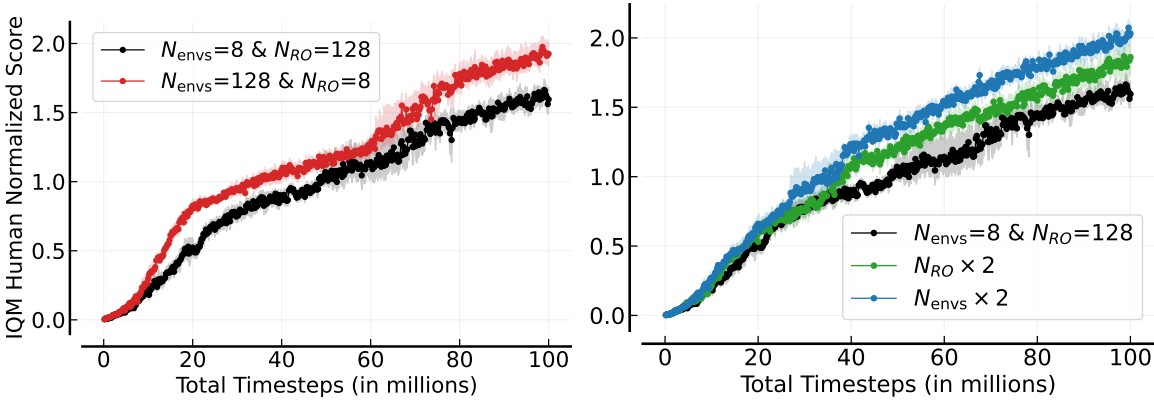

*Figure 14.* **Evaluating the impact of scaling parallel data collection** for PQN agent (Gallici et al., 2024) on Atari-10 benchmark (Aitchison et al., 2023). See Section 4.1 for training details.

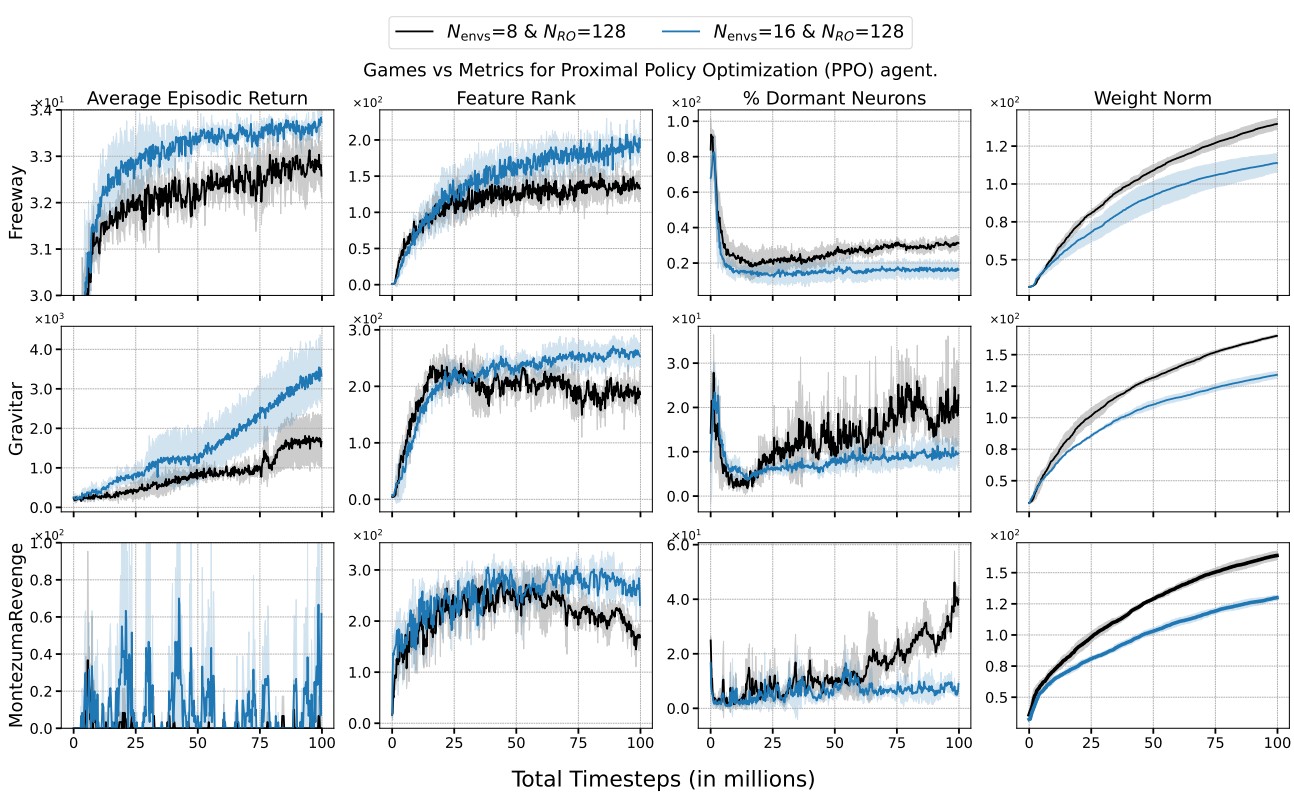

*Figure 15.* **Evaluating the impact of increasing $N_{\text{envs}}$ on a separate set of ALE games.** These are the so-called "hard exploration games" from Taiga et al. (2020). We can observe that increased batch size results in equivalent or improved performance, and overall improvement on the learning dynamics measures. All results averaged over 5 seeds, shaded areas represent 95% confidence intervals. See Section 4 for training details.

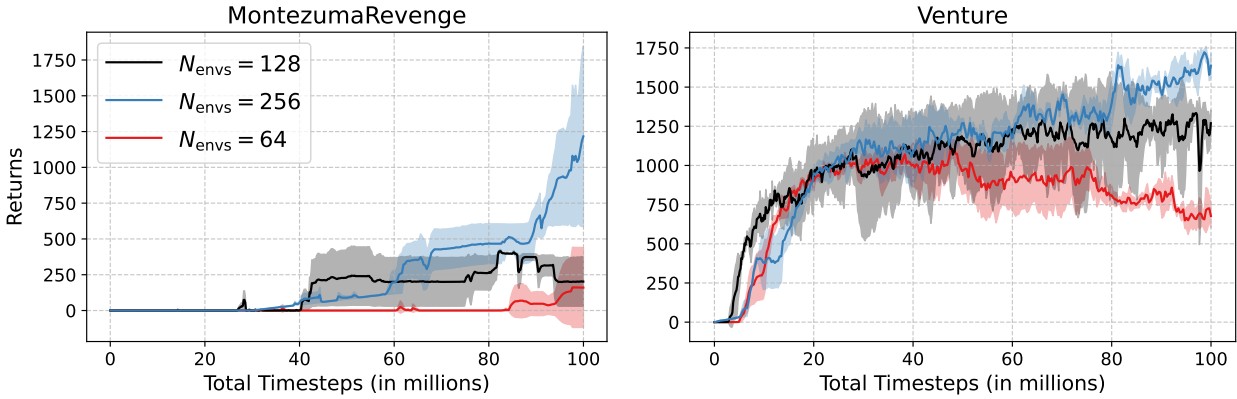

*Figure 16.* **Evaluating the impact of increasing $N_{\text{envs}}$ on a separate set of ALE games.** These are the so-called "hard exploration games" from Taiga et al. (2020). Scaling data by increasing the $N_{\text{envs}}$ is more effective than $N_{RO}$. All results averaged over 5 seeds, shaded areas represent 95% confidence intervals. For this experiment, we use PPO+RND (Burda et al., 2019) algorithm from Cleanrl (Huang et al., 2022) library.

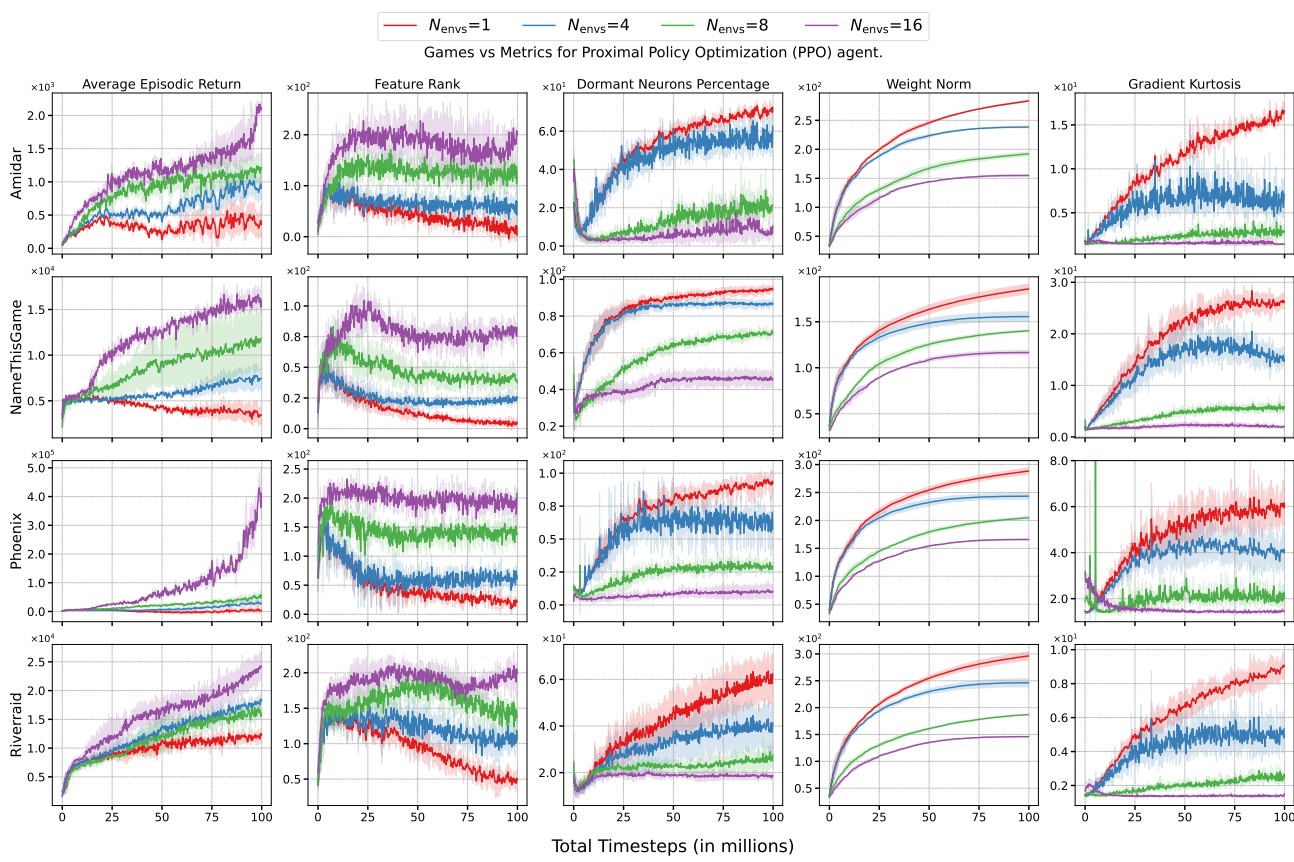

*Figure 17.* **Empirical analyses for four representative games with different amount of parallel data for PPO (Schulman et al., 2017).** From left to right: training returns, feature rank, dormant neurons percentage (Sokar et al., 2023), weight norm and gradient kurtosis. Increasing $N_{\text{envs}}$ for parallel data collection improves final performance. All results averaged over 5 seeds, shaded areas represent 95% confidence intervals.

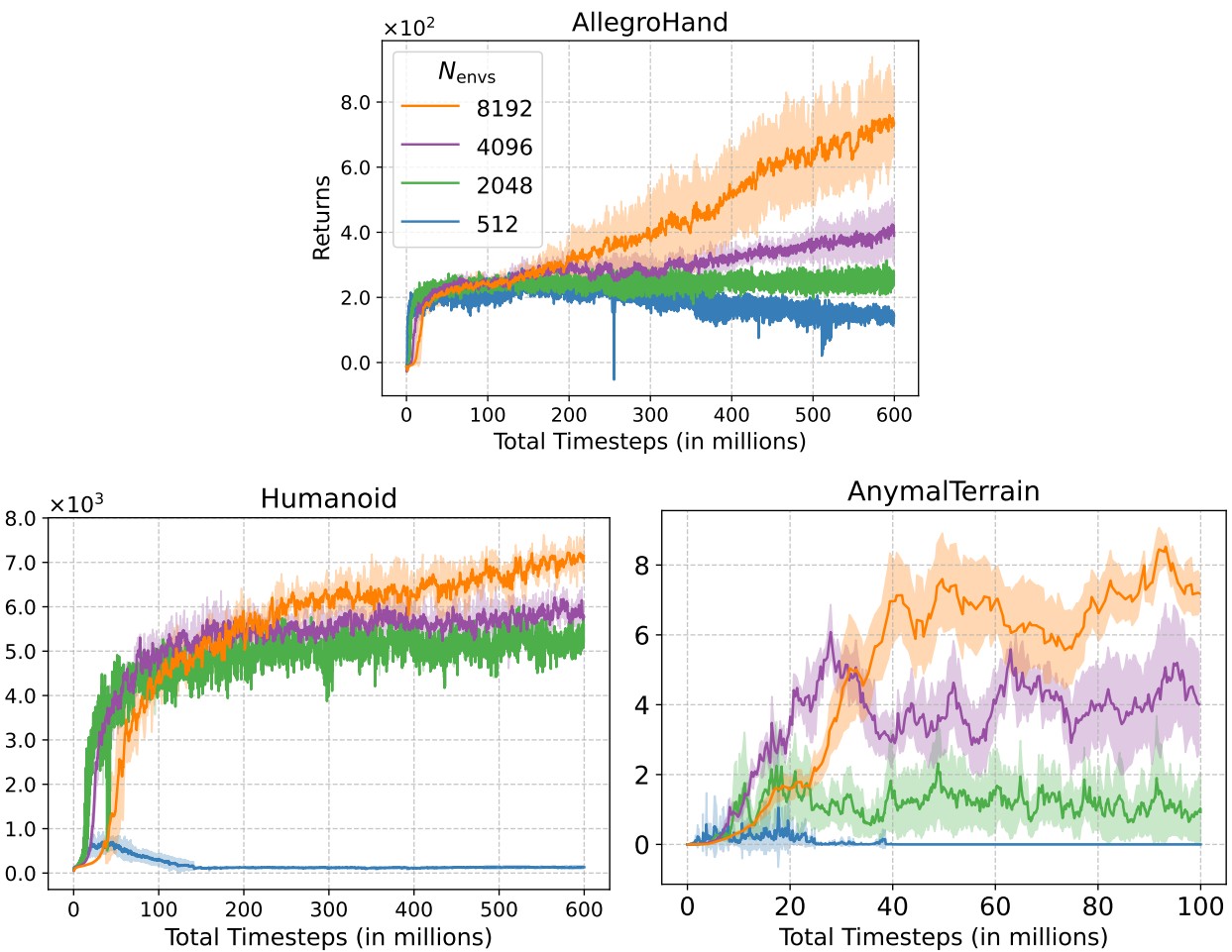

*Figure 18.* **Proximal Policy Optimization (PPO)** (Schulman et al., 2017; Huang et al., 2022) on Isaac Gym environments (Makoviychuk et al.) when increasing $N_{\text{envs}}$. Increasing $N_{\text{envs}}$ for parallel data collection improves final performance. We report returns over 5 runs for each experiment.

