# OpenReview forum: "The Impact of On-Policy Parallelized Data Collection on Deep Reinforcement Learning Networks"
_ICML.cc/2025/Conference — ICML 2025 poster_

### Official Review · Reviewer_NUj3 · 2025-02-20

**Overall Recommendation:** 3

**Summary:**

The paper investigates how scaling parallel data collection (i.e., the product of the number of parallel environments
$N_\text{envs}$ and rollout length ($N_\text{RO}$) affects the performance and representation quality of deep RL agents, focusing primarily on PPO (and briefly on a value-based variant, PQN).

The authors shop empirically, that for the same batch size $B$ using a larger $N_\text{envs}$ yields better learning performance on Atari-10. They attempt to correlate this finding with known metrics that are linked to loss of plasticity showing that larger $N_\text{envs}$ appear better in this regard.

**Claims And Evidence:**

> Claim 1: keeping the total batch size fixed $B$ fixed, more parallel environments yield superior performance compared to longer rollouts in fewer environments.

While the experiments that the authors provide can suggest this, they are only performed on two different configurations ($N_\text{envs}=8$, $N_\text{RO} = 128$ and $N_\text{envs}=128$, $N_\text{RO} = 8$).
The claim is too general and suggests that there should be a trend (potentially a scaling law), but in order to warrant such a claim there would need to be experiments for different sets of parameter configurations for $N_\text{envs}$ and $N_\text{RO}$.
Currently, all that can be said is that the one hyperparameter configuration beats the other one. we do not know if e.g. $N_\text{envs}=32$, $N_\text{RO} = 32$ would be the best one for example.

> Claim 2: Scaling the total collected data can mitigate issues such as loss of plasticity.

While the authors provide some metrics that have been linked to loss of plasticity, I am unsure whether their claim is justified here. First, prior work has used to explain potential reasons for the loss of plasticity, however, simply because the fraction of dormant neurons or the weight norms are higher, does not mean that plasticity has been lost. Especially, since the agents' performance still increases, i.e. there might be a correlation here, however, it's not clear if it is causal and if loss of plasticity is happening.

**Essential References Not Discussed:**

-

**Experimental Designs Or Analyses:**

see above.

**Methods And Evaluation Criteria:**

I am not sure, why the authors chose the combination of PPO and Atari, i.e. an algorithm for contrinuous action environments on discrete action environments for their main analysis. Especially as PQN (which as a DQN/value-based method is the more common choice for Atari) does not show any difference in performance. The authors claim PQN still shows improved performance in the "learning dynamics metrics" they track, however, I am unsure about the significance here.

As mentioned above, in order to convincingly show scaling behaviour, the authors need to provide more probing points for different configurations and show some trend.

Further, the paper would also benefit if in addition the same behaviour could be shown on continuous control tasks. Especially, since there have been works on massively scaling PPO there (Rudin et al. Learning to Walk in Minutes Using Massively Parallel Deep Reinforcement Learning, 2021).

**Other Comments Or Suggestions:**

- The paper does not have a seperate conclusion section. This should be added as currently, the discussion also contrains some conclusions. The author should follow common practice to seperate this into an independent section.
- A separate analysis for long-horizon tasks (or tasks with sparse reward signals) could highlight whether shorter rollouts are indeed always better or if certain environments require longer rollouts to reduce bias.

**Other Strengths And Weaknesses:**

Strength:
-

Weaknesses:
- Scope is limited to few discrete action environments
- Too few parameter configurations in order to make a general claim about the scalability
- Some experimental details are missing which should be added (e.g. batch size)
- No theoretical analysis or insights

**Questions For Authors:**

Have the authors tested or do they plan to test beyond, say, 256 or 512 parallel environments to see if the improvements continue (or saturate)? Especially given that there have been works in continuous state-action RL (Rudin et al. 2021).

**Relation To Broader Scientific Literature:**

-

**Theoretical Claims:**

The authors make no theoretical claims in the paper.

---

> ### Author Rebuttal · Authors · 2025-04-01
>
> We thank the reviewer for their feedback, useful comments, and address their concerns below.
>
> ## The claim is too general and suggests that there should be a trend (potentially a scaling law)
>
> We ran extra experiments varying the number of environments, rollout lengths, and across different domains to support the claims made in the paper. Please [see Figures 1,2,3,4,5](https://shorturl.at/Ycvgx). Figs 1, 2, and 5 show that increasing #Env improves final performance. Additionally, Fig 1 demonstrates that representation collapse and the percentage of dormant neurons are mitigated when using a larger #Env. Figs 3 & 4 confirm that increasing #Env is more effective than increasing #Rollouts. Notably, Fig 4 also shows performance boosts in sparse reward games.
>
> ##  higher dormant neurons does not mean plasticity has been lost
>
> We agree that increased dormancy or weight norm changes do not directly imply a loss of plasticity. We were motivated to investigate these metrics given their use as a proxy for plasticity loss in recent papers [1,2,3,4,5].  Our intent was to show a correlation between these features and training dynamics, not to assert causality.
>
> ## why PPO and Atari?
>
> While PPO was originally proposed for continuous control, it has been widely and successfully used in discrete action settings, especially in the Atari domain [6]. We chose PPO due to its popularity, and relevance in benchmarking studies and studying deep RL learning dynamics [1]. Additionally, we chose ALE, as it  is a well known benchmark for studying issues such as loss of plasticity [7,8], so using it provides a useful reference point for our findings.
>
> Nonetheless, we have added extra experiments on IsaacGym [9] which confirm our findings (see [Figures 2,3,5](https://shorturl.at/Ycvgx)).
>
>
> ## unsure about the significance of PQN “learning dynamics metrics” improvements
>
> While we agree that the performance improvements with PQN are not as pronounced, our findings do suggest that using a larger number of environments leads to more parameter efficient learning. Nevertheless, we will soften our claims on PQN to avoid misrepresenting our results.
>
> ## can same behaviour be shown on continuous control tasks?
>
> We ran experiments on Isaac Gym [9], and the results support the main thesis of the paper. Larger #Envs mitigate optimization issues and improve performance, [see Figs 2,3,5](https://shorturl.at/Ycvgx).
>
> ## seperate conclusion section
>
> We will restructure the paper to include a dedicated conclusion section.
>
> ## separate analysis for long-horizon tasks (or tasks with sparse reward signals)...
>
> While our current set of tasks includes varying levels of difficulty and reward sparsity, we agree that explicitly studying sparse-reward environments would provide deeper insight. To this end, we examine the impact of on-policy parallelized data collection on two  hard exploration [10] games using PPO+RND [11] from CleanRL [12]; [see Figure 4](https://shorturl.at/Ycvgx). We observe that the claims made in the paper generally hold in these scenarios as well.
>
> ## test beyond 256 or 512 parallel environments to see if the improvements continue
>
> While we have begun running these experiments, they will not be able to finish before the rebuttal period as they are more computationally expensive.  Once completed, we plan to include larger-scale experiments in the final version.
>
> However, the new IsaacGym results suggest that our results hold when scaling to higher values ([see Figures 2, 3,5](https://shorturl.at/Ycvgx)).
>
> ## References
>
> [1] Moalla et al. No representation, no trust: Connecting representation, collapse, and trust issues in PPO. NeurIPS’24.
>
> [2] Juliani & Ash. A Study of Plasticity Loss in On-Policy Deep Reinforcement Learning. NeurIPS’24.
>
> [3]. Ahn et al. "Prevalence of Negative Transfer in Continual Reinforcement Learning: Analyses and a Simple Baseline." ICLR’25.
>
> [4]. Nauman et al. "Overestimation, Overfitting, and Plasticity in Actor-Critic: the Bitter Lesson of Reinforcement Learning." ICML’24.
>
> [5]. Dohare et al. "Loss of plasticity in deep continual learning." Nature (2024)
>
> [6]. Ellis et al.  Adam on Local Time: Addressing Nonstationarity in RL with Relative Adam Timesteps. NeurIPS’24.
>
> [7]. Nikishin et al. "Deep reinforcement learning with plasticity injection." NeurIPS’23
>
> [8] Lyle et al. "Understanding plasticity in neural networks." ICML’23
>
> [9]. Makoviychuk et al. Isaac gym: High performance gpu based physics simulation for robot learning. 2021
>
> [10]. Taiga et al. "On Bonus Based Exploration Methods In The Arcade Learning Environment." ICLR’20
>
> [11]. Burda et al. Exploration by random network distillation. ICLR’19
>
> [12]. Huang et al. "Cleanrl: High-quality single-file implementations of deep reinforcement learning algorithms." JMLR’22

---

### Official Review · Reviewer_qnDC · 2025-03-12

**Overall Recommendation:** 3

**Summary:**

This paper focuses on the problem of reinforcement learning with multiple environments, which has gained increasing interest over the past years due to GPU utilization. Through empirical analysis of the effect of the number of environments and the length of rollouts, the authors provide recommendations (e.g., increasing the number of parallel environments) on how to improve the performance of deep RL agents under this parallel environment setting.

**Claims And Evidence:**

The authors tend throughout the paper to claim a relationship between two variables using two data points, which is unsubstantiated.
- For example, one of the main claims in the paper is that increasing the number of environments yields more gains with or without fixing the update budget (Figure 1a, 1b). To claim such a trend, the authors need to show the performance on the y-axis and different values of # environments on the x-axis. That is, the performance increases when using x2/x3/x4/etc the number of environments. The claim can be made if we can observe a positive correlation between the two variables. Additionally, it would be even better if the authors could show this for multiple update budgets.
- Another example is where the authors claim a positive correlation between $N_{\text{envs}}$ and performance/feature-rank. Also, they claim a negative correlation between $N_{\text{envs}}$ and the level of neuron-dormancy/weight norm. Those conclusions are followed by experiments on PQN, which mostly contradicts what the authors discovered with PPO. The results with PQN should allow the authors to reconcile their claims and never overstate the conclusions.
- The results with PQN seem to be contradictory to the results from the PQN paper. The authors here showed that increasing the number of environments doesn’t help, whereas the PQN paper shows it does. The authors need to explain this discrepancy.
- In Figure 5, The authors claimed that increasing $N_{\text{envs}}$ in PQN mitigates representation deterioration and slightly improves final performance, but the results are inconclusive.

**Essential References Not Discussed:**

No

**Experimental Designs Or Analyses:**

The main issue in the evaluation is the number of independent runs (3 runs) is very low, so I suspect the statistical significance of the results. I highly encourage the authors to increase the number of runs to at least 10.

**A minor issue:** the authors need to show the dependent variable on the y-axis and the name of the environment as the title, not the other way around.

**Methods And Evaluation Criteria:**

The authors evaluate the Atari arcade environments, which is a well-known benchmarking suite and suitable for the scope of the paper and its parallel environment setting.

**Other Comments Or Suggestions:**

In line 92, value-based (McKenzie & McDonnell, 2022) -> this reference needs to be updated with an older reference.

**Other Strengths And Weaknesses:**

The study is well-timed since there is an increasing interest in using RL under the parallel environment setting, which is parallelizable with modern GPUs. Usually, parallelizable methods such as PQN/PPO are presented with a specific set of hyperparameters, so it is nice to have a paper dedicated to studying this setting and finding best practices that apply to this category of methods.

I have concerns about the experiments and the validity of the conclusions. Please refer to my comments under the claims and evidence section.

**Questions For Authors:**

To have a fair comparison, I want to make sure that the number of total environment interactions ($N_{\text{envs}}\times T$, where T is the number of interactions in each environment) is the same when comparing algorithms that use different $N_{\text{envs}}$. Can the authors confirm that this is the case?

**Relation To Broader Scientific Literature:**

The related works section covers prior work.

**Theoretical Claims:**

No theoretical claims are presented in the paper.

---

> ### Author Rebuttal · Authors · 2025-04-01
>
> We thank the reviewer for their feedback! We are glad that the reviewer finds that “the study is well-timed”, and “nice to have a paper dedicated to studying this setting and finding best practices”.
>
> We address their main concerns below.
>
> ## two variables using two data points, which is unsubstantiated.  To claim such a trend, the authors need to show the performance on the y-axis and different values of # environments on the x-axis. That is, the performance increases when using x2/x3/x4/etc the number of environments.
>
> We agree with the reviewer that more data points are needed to support this claim. We conducted additional experiments by varying the number of environments ([see Figs 1,2,3](https://shorturl.at/Ycvgx)); following your suggestion, we also plot the number of environments (#Envs) on the x-axis against returns ([see Fig 5](https://shorturl.at/Ycvgx)) and agree that this helps clarify our message. The conclusions remain the same: increasing the number of environments mitigates optimization-related issues in deep RL and improves performance, unlike simply increasing the number of rollouts.
>
>
> ## Another example is where the authors claim a positive correlation between  and performance/feature-rank.
>
>
> To support our claims, we report training returns, feature rank, percentage of dormant neurons, weight norm, and gradient kurtosis for different values of #Envs. ([see Figure 1](https://shorturl.at/Ycvgx)) shows a strong correlation between #Envs and the metrics used to study the dynamics of deep RL agents [1, 2]. Larger #Envs mitigate feature collapse and reduce dormant neuron percentage.
>
> ## contradictory results with PQN, discrepancy with original paper, and inconclusive results
>
> We would like to highlight that the focus of our paper is on PPO. PQN was chosen solely to conduct a comparative analysis with PPO, as both algorithms use on-policy parallelized data collection but differ in their loss functions. We will soften our claims to avoid overstating the findings.
>
> The discrepancy with the published results likely arises from differences in the experimental framework and hyperparameter tuning. Given that we are studying PQN strictly in reference to PPO, we chose to use the same set of hyperparameters across both algorithms to isolate the effects of the loss function and data collection strategy. Notably, while PPO uses 4 epochs by default, the authors of PQN used 2, which results in improved performance. We will clarify these differences in the final version.
>
> We realize that perhaps including PQN in the background section gives the impression that PQN is part of the main focus of the paper, and propose to move section 2.2 to the appendix to avoid this confusion.
>
>
> ## number of independent runs (3 runs) is very low
>
>
> The 3 runs specified in Figures 7 and 10 are incorrect, the actual number of runs were 5; we apologize for the confusion and will correct the typo. We followed the experimental setup from [1,2], where they run experiments with 5 seeds.  Further, we are running additional experiments with 5 more seeds to increase statistical significance and will include them in the final version of the paper.
>
> ##  is the number of total environment interactions the same when comparing algorithms?
>
> The number of environment interactions is in fact not the same when comparing with different batch sizes (whether by changing the number of environments or the rollout length). PPO (and PQN) leverage parallelism via simulated environments for their training setup, and it is this setting we explore in our work. As such, we compare the various methods across gradient steps as opposed to environment interactions, as this is a better indication of how well parallelized data collection can be leveraged for learning. Thank you for raising this, as it is a subtle point that we will clarify in our final version.
>
>
> ## suggestions for figure clarity and citation update
>
>
> Thank you for raising these points, we will correct them.
>
>
> ## References
>
>
> [1]. Skander Moalla, Andrea Miele, Daniil Pyatko, Razvan Pascanu, and Caglar Gulcehre. No representa- tion, no trust: Connecting representation, collapse, and trust issues in PPO. NeurIPS’24.
>
> [2]. Obando-Ceron, J., Sokar, G., Willi, T., Lyle, C., Farebrother, J., Foerster, J. N., Dziugaite, G., Precup, D., and Castro, P. S.. Mixtures of experts unlock parameter scal- ing for deep rl. ICML’24

---

> > ### Comment · Reviewer_qnDC · 2025-04-08
> >
> > I would like to thank the authors for their response. It addresses most of my concerns. The remaining issue is that the total number of environment interactions vary across different baselines. This makes the evaluation unfair. The total number of interactions has to be fixed to have a fair comparison. The data collection scheme can differ and this tells us which data scheme is better. For example, better performance with increasing the number of environment might be only because the total number of interactions (samples) increased which is not surprising since the performance in most algorithms scale with number of samples.
> >
> > Can the authors provide results for the case when the total number of environment interactions is fixed across all variations?

---

> > > ### Author Response · Authors · 2025-04-08
> > >
> > > We thank the reviewer for reading our responses and are glad that they addressed most of your concerns.
> > >
> > > With your latest comments we realized that we originally misunderstood your question *“Is the number of total environment interactions the same when comparing algorithms?”*. After careful inspection we would like to clarify that **the number of environment interactions is fixed across all experiments when varying  $N_{\textrm{envs}}$ in all of our figures.** We provide more details below.
> > >
> > > Our work examines the impact of parallel data collection by varying $N_{\textrm{envs}}$ and $N_{\textrm{RO}}$. Changes in these values result in a different number of environment steps _per iteration_, as per the CleanRL [1] codebase. Specifically:
> > >
> > >   -`batch_size` is calculated as: $N_{\textrm{envs}} \times N_{\textrm{RO}}$
> > >
> > >   -`num_iterations` is calculated as:  $\left\lfloor\frac{\textrm{totalTimesteps}}{\textrm{batchSize}}\right\rfloor$
> > >
> > > Based on the structure of the CleanRL codebase, we set the `total_timesteps`, which is equivalent to the total number of environment steps. This means that **the number of environment interactions is fixed across all experiments when varying  $N_{\textrm{envs}}$ in all of our figures.**
> > >
> > > The following table illustrates how these values are computed, and we link each term to its corresponding line of code to provide evidence of what was actually run. Again, note that `total_timesteps` is equivalent to total number of environment interactions, and **equal for all settings considered**.
> > >
> > > | **total_timesteps ([see code](https://shorturl.at/ewgC4))**     | **100M**   | **100M**   | **100M**   | **100M**   | **100M**   |
> > > |:-----------------------:|:----------:|:----------:|:----------:|:----------:|:----------:|
> > > | **num_envs (Nₑₙᵥₛ, [see code](https://shorturl.at/NquVC))**            |     1      |     2      |     4      |     8      |    16      |
> > > | **num_steps (Nᵣₒ, [see code](https://shorturl.at/eezJ0))**           |   128      |   128      |   128      |   128      |   128      |
> > > | **batch_size ([see code](https://shorturl.at/FKlSK))**          |   128      |   256      |   512      |   1024     |   2048     |
> > > | **num_iterations ([see code](https://shorturl.at/IpvhI))**      | 781250     | 390625     | 195312 | 97656 | 48828 |
> > >
> > > Finally, we realize that we incorrectly used "Number of Iterations" as the x-label in a few of our figures, when it should read "Total Timesteps" (as in Figure 1), which may have possibly led to the confusion. This was simply a typo on our axis labeling, but importantly _the tick marks represent timesteps, not iterations_, consistent with our response above.
> > >
> > >
> > > Thank you very much for pressing us on this point, as it is crucial to have clarity on. To reiterate: **the performance boost is not due to having more interactions with the environment, as the number of environment steps is fixed across all experiments**.
> > >
> > > We hope this response addresses your remaining concern, and if so encourage you to revise your score. Thank you again for all the valuable feedback, and engagement with us!
> > >
> > > [1]. Huang et al. "Cleanrl: High-quality single-file implementations of deep reinforcement learning algorithms." JMLR’22

---

### Official Review · Reviewer_ZCcE · 2025-03-12

**Overall Recommendation:** 4

**Summary:**

The paper claims that larger batch sizes in deep reinforcement learning obtained by parallelized data collectors help mitigate several optimization challenges, listed below, and give recommendations on the scaling of the dimensions of the batch size (num envs vs rollout size).
1) Performance gains: Increasing the batch size improves sample efficiency of PPO and PQN.
2) Batch dimensions recommendation: It is better to increase the number of parallel environments than the rollout size.
3) Better learning dynamics:
  a) a higher batch size improves plasticity and learned representations (higher feature rank, less dormant neurons, lower weight norm, lower kurtosis, less policy variance, higher effective sample size)
  b) A higher batch can be more important when using separate actor and critic networks.
4) Connection to other hyperprameters: it seems unnecessary to vary the learning rate and discount factor when varying the dimensions of the batch size.

## update after rebuttal
I have read the other reviews and comments from the authors, and I maintain the score of my review. The authors agreed to make their contributions clearer and explained the validity of their experimental protocol, which I encourage them to make much clearer in the final version of the paper.

**Claims And Evidence:**

All the claims are backed by sufficient empirical evidence.

1) Figure 1
2) Figure 1
3a) Figure 2, Figure 3
3b) Figure 4
4) Appendix Figure and Figure 9.

Claims 1, 2 are well illustrated by using aggregated performance metrics across the 10 envs, however, claims 3b and 4, which also argue about performance, only use individual env curves, which could be prone to cherry-picking. While it may be the case that the paper wanted to present more details by including individual envs, I believe that these claims would be better illustrated (have a complete point) by using the aggregate performance over the 10 envs like done in Figure 1.

**Essential References Not Discussed:**

The essential references are discussed.

**Experimental Designs Or Analyses:**

The paper makes a good use of aggregate metrics (Figure 1) when claiming overall performance improvement and using individual env training curves (Figures 2, 3, 4, 5) to show improvements on learning dynamics, which cannot easily be aggregated.
Nevertheless, I have made comments in the section about claims on how I believe the paper could make better use of these two reporting strategies.

The choice of the number of environments and seeds makes a good balance for statistical significance.

The paper uses a safe and sound experimental protocol by using an implementation by a popular codebase (CleanRL) with its default hyperprameters.

**Methods And Evaluation Criteria:**

The Atari benchmark with the subset of Atari-10 is a good choice for this work, as it's been widely used in similar work on plasticity.
The work does not need to compare to other methods to make its claim; a comparison with the base PPO is enough.

**Other Comments Or Suggestions:**

No extra comments.

**Other Strengths And Weaknesses:**

Presentation & clarity

It's not easy to identify the precise main claims of the paper from the abstract and the introduction. Both of these sections present a global claim that a higher batch size through parallelized data collectors helps mitigate several optimization challenges, but nothing more precise.
The reader has to wait for the conclusion of each section to get the precise claim. I believe the paper would be better if the claims were presented in a precise way earlier in the paper.

Impact

The claims made in the paper are interesting and give important insights to RL researchers and practitioners, however, they are quite narrow as the benefit of increasing the batch size comes with induced extra compute, and the paper does not compare to other methods that could use this extra compute more effectively.

**Questions For Authors:**

No additional questions.

**Relation To Broader Scientific Literature:**

The paper related its finding to the plasticity loss literature, but it is likely that other optimization features are impacted by varying the batch size, the paper does not discuss such broader relations.

**Theoretical Claims:**

The paper does not make any theoretical claim.

---

> ### Author Rebuttal · Authors · 2025-04-01
>
> We thank the reviewer for their feedback! We are happy that the reviewer found that “the claims made in the paper are interesting and give important insights to RL researchers and practitioners” and that “the claims are backed by sufficient empirical evidence”.
>
> We respond to their main concerns below.
>
> ## Only use individual env curves, which could be prone to cherry-picking.
>
> Due to our limited compute budget, we reported individual game curves only for the subset of games included in the main analysis of the paper. However, we will run the experiments for the remaining games in the suite and share the corresponding aggregated plots in the final version of the paper. Note that Phoenix and NameThisGame were the two games used for studying representation collapse on PPO [1].
>
>
> [1]. Skander Moalla, Andrea Miele, Daniil Pyatko, Razvan Pascanu, and Caglar Gulcehre. No representa- tion, no trust: Connecting representation, collapse, and trust issues in PPO. In Advances in Neural Information Processing Systems, 2024.
>
>
> ## It's not easy to identify the precise main claims of the paper from the abstract and the introduction.
>
>
> Thank you for the suggestion regarding presentation and clarity. We will include precise statements of our claims in both the abstract and the introduction in the final version.
>
>
> ## … the benefit of increasing the batch size comes with induced extra compute, and the paper does not compare to other methods that could use this extra compute more effectively
>
>
> We agree that increasing batch size comes with additional computational cost, and we acknowledge that our current comparisons focus primarily on data collection strategies within a fixed algorithmic framework (mainly PPO). Our intention with this study is to isolate and understand the trade-offs inherent in data collection design, specifically, how varying the number of parallel environments and rollout lengths affects learning dynamics and final performance. In the final version, we will clarify this scope more explicitly and emphasize that our results are complementary to other lines of work focusing on algorithmic improvements or more compute-efficient approaches.

---

> > ### Comment · Reviewer_ZCcE · 2025-04-04
> >
> > After looking at the other reviews and the comments from the authors, I acknowledge some of the limitations mentioned by the other reviewers, especially the lack of depth in the claims made with PQN (which is now treated as an on-policy algorithm?).
> > I maintain the score of my review but strongly encourage the authors to make the claims about PQN more precise and limited to the results in their figures.

---

> > > ### Author Response · Authors · 2025-04-07
> > >
> > > We thank the reviewer for reading our responses and the feedback from the other reviewers. We will revise the manuscript to ensure that our claims about PQN are carefully stated and strictly supported by the empirical results shown in the figures.
> > >
> > > Best regards,
> > >
> > > Authors, Paper 7853

---

### Decision · Program_Chairs · 2025-05-01

**Decision:**

Accept (poster)

**Comment:**

This paper investigates the batch size problem in parallel on-policy RL and demonstrates that scaling the number of environments offers several benefits, such as performance enhancement and improved representation.

Although the paper's scope appears narrow, it presents extensive experiments across various environments (both discrete and continuous) and hyperparameter settings. All reviewers concur that the claims made in the paper are intriguing and provide valuable insights for RL researchers and practitioners.